# Length Representations in Large Language Models

## Abstract

Large language models (LLMs) have shown remarkable capabilities across various tasks, that are learned from massive amounts of text-based data. Although LLMs can control output sequence length, particularly through instruction-based settings, the internal mechanisms behind this control has been unexplored. In this study, we provide empirical evidence on how output sequence length information is encoded within the internal representations of LLMs. In particular, our findings show that multi-head attention mechanisms are critical in determining output sequence length, which can be adjusted in a disentangling manner. By scaling specific hidden units within the model, we can control the output sequence length without losing the informativeness of the generated text, thereby indicating that length information is partially disentangled from semantic information. Moreover, some hidden units become increasingly active as prompts become more length-specific, thus reflecting the model's internal awareness of this attribute. Our findings suggest that LLMs have learned robust and adaptable internal mechanisms for controlling output length without external controls.[1]

## 1 Introduction

Large language models (LLMs) have gained considerable attention in recent years for their remarkable task-solving capabilities (Ouyang et al., 2022; Wei et al., 2022; Bubeck et al., 2023). LLMs are trained to predict the next word in a sequence. They can produce coherent and informative text, which demonstrates their implicit understanding of diverse linguistic structures (Tenney et al., 2019; Niu et al., 2022; Beguš et al., 2023). Furthermore, they also learn when to stop generating text to ensure that the output adheres to appropriate length constraints (Juseon-Do et al., 2024). In LLMs, controlling output sequence length is crucial for real-world applications, such as summarization, where fitting content within specified length limits without losing informativeness is crucial. Therefore, the number of studies attempting to improve length controllability has increased drastically (Shen et al., 2023; Jie et al., 2024; Yuan et al., 2024).

Based on advancements in instruction-based LLMs, it is observed that injecting constraints into prompts can further effectively control output length without requiring model modifications (Juseon-Do et al., 2024). However, these prompt engineering methods mainly focus on external controls, and it has not been explored how LLMs internally encode and constrain output sequence length. Users of LLMs usually have a desired length for generated texts in applications, such as text summarization (Liu et al., 2018; Makino et al., 2019; Liu et al., 2022; Kwon et al., 2023), machine translation (Wu et al., 2016; Murray & Chiang, 2018; Zhuocheng et al., 2023), knowledge QA, or dialogue generation (Liu et al., 2020; Gupta et al., 2021). Understanding these internal mechanisms is critical for achieving precise length control, while enhancing the interpretability and robustness of LLMs in generation-based systems. Herein, we aim to investigate how output sequence length information is encoded within the internal representations of general transformer architectures. Specifically, we first investigate which components within LLM transformer layers contribute to length control. Our findings reveal that the outputs from multi-head attention mechanisms in the lower layers play a key role in determining and controlling output sequence length in a tunable and disentangled manner.

We empirically demonstrate, based on human evaluations, that we can adjust output length during generation without losing the coherence and informativeness of texts by scaling specific hidden

---

[1]Our code is available at `https://github.com/XXXX`.

units within the outputs from the lower layers of multi-head attention mechanisms. For instance, multiplying certain hidden units with negative numbers results in longer text, while multiplying them by positive numbers generates more concise texts without losing informativeness. Furthermore, certain hidden units related to length information show increased activity as prompts become more specific regarding length constraints. These units appear to be directly involved in controlling output length, indicating that LLMs have learned to process length-related information as a distinct feature, partially disentangled from other semantic information. Moreover, we find that the same highly activated hidden units are consistently involved in length control even after fine-tuning, regardless of length constraints in prompts (Dai et al., 2023).

For this, we utilize a sentence summarization task, which often requires adherence to desired summary lengths. In this study, we employ models from the Llama-2 family, including Llama-3-8B,and the Phi-3 family.

## 2 RELATED WORK

**Large Language Models.** In recent years, LLMs have achieved considerable success due to their remarkable task-solving abilities, specifically in zero-shot settings (Radford et al., 2019; Brown et al., 2020). LLMs have been broadly categorized into open and closed models. The open models, such as the Llama or Phi family, offer flexible access to modify their architectures, while the closed models, such as ChatGPT,[2] have demonstrated remarkable reasoning abilities in various natural language processing tasks (Jiao et al., 2023; Peng et al., 2023; Laskar et al., 2023; Ye et al., 2023; Xie et al., 2023; 2024; Juseon-Do et al., 2024). Recent studies have focused on finding better methods to prompt LLMs (Zhou et al., 2022; Kojima et al., 2023; Zhou et al., 2023).

**Interpretability.** Due to increasing interest in investigating the internal mechanisms of deep neural networks (Räuker et al., 2023), significant attempts have been made to understand LLMs with a focus on models like BERT (Tenney et al., 2019; Rogers et al., 2020; Niu et al., 2022), GPT (Hanna et al., 2023), and even multimodal models (Goh et al., 2021). For instance, Gurnee & Tegmark (2024) showed that, when handling various prompts, LLMs learn linear representations of space and time across multiple scales that show robustness. They also showed that next token prediction can be changed simply by disentangling hidden units related to time. Heinzerling & Inui (2024) introduced directions that encode numeric properties in an interpretable manner; hence, by disentangling these representations, LLM prediction can change accordingly. Moreover, there have been attempts to investigate how in-context learning with LLMs behaves similar to explicit fine-tuning for better understanding them (Dai et al., 2023). Early efforts to investigate how neural networks treat length information have focused on memory cell networks in LSTMs, as they recursively encode and decode sequences, though they failed to find single units related to length information (Shi et al., 2016).

**Length Controllable Summarization.** Text summarization aims to produce a concise summary from an original text by retaining informative contents (Liu et al., 2018; Takase & Okazaki, 2019; Li et al., 2020; He et al., 2022). As the summarization often requires additional constraints such as a desired summary length, previous studies have focused on learning length-specific parameters (Kikuchi et al., 2016; Schumann et al., 2020; Ghalandari et al., 2022), injecting direct constraints (Takase & Okazaki, 2019; Makino et al., 2019), or splitting the training dataset into specific length ranges (He et al., 2022). Recently, Juseon-Do et al. (2024) considered in-context learning and demonstrated that LLMs can control output sequence length through "length priming". This method involves injecting more length-specific information into prompts, thereby allowing the model to adjust output sequence length without modifying model architectures or learning parameters. Jie et al. (2024) considered length control types such as greater/smaller than a value with exhaustive model modifications with reinforcement learning.

To the best of our knowledge, this study is the first attempt to interpret how length information is encoded in LLMs and demonstrate how length-specific information is partially disentangled from semantic information. Furthermore, by comparing various length-specific prompts, we investigate how in-context learning and fine-tuning can influence the internal representations of LLMs with

---

[2]https://chat.openai.com/

different prompts. Finally, we demonstrate how disentangling length-specific hidden units can adjust output sequence length without losing informativeness.

## 3 Finding Length Representations

Our goal is to understand whether and how length representations are encoded in LLMs when using various length-constraint prompts. For this, we extracted outputs from different components and layers of transformer architectures during text generation. We then applied linear regression to predict the generation time steps from these hidden states. We used the coefficient of determination, $R^2$, to evaluate the extent to which internal states of transformer capture length representations.

### 3.1 Summarization Dataset

For our investigation, we used the **Google** sentence summarization dataset,[3] given that summarization often requires additional constraints, such as a desired summary length (Takase & Okazaki, 2019; Schumann et al., 2020; Ghalandari et al., 2022). This dataset was automatically created by considering the syntactic dependency tree structure from news headlines (Filippova & Altun, 2013). The training, validation, and test datasets consist of 200,000, 1,000, and 1,000 pairs, respectively. The average gold compression ratio is 0.45 for the test dataset used in the evaluation.

We used the dataset in an instruction-based format following previous work (Juseon-Do et al., 2024).[4] Table 1 presents the instruction templates. As can be seen, in the **No-constraint** setting, the model summarizes a given sentence without considering a desired length, while in the **Length** setting, it summarizes the sentence with a specific desired length. The **Priming** setting further considers more specific length information, such as the length of a given sentence and the number of words to keep. We inject the length of ground-truth summaries for the length constraint instructions.

Table 1: Instruction formats for length constraints. "src" indicates the placeholder for a source sentence, "del" denotes the placeholder for the number of deleted words, and "keep" and "src len" denote additional length information.

| Constraint | Instruction |
|---|---|
| No-constraint | Sentence:\n{src}\nThe sentence without the less important words would be:\n |
| Length | Sentence:\n{src}\nThe sentence without the less important {del} words would be:\n |
| Priming | Sentence that consists of {src len} words:\n{src}\nThe sentence that consists of {keep} words without the less important {del} words would be:\n |

### 3.2 Models and Methods

**Models.** We performed our experiments using the Llama-2 family of pre-trained LLMs, which range from 7B to 70B parameters (Touvron et al., 2023). These include the Llama-3-8B (AI@Meta, 2024) and the Phi-3 family of Phi-3-mini-4k-instruct, as well as the Phi-3-small-8k-instruct (Abdin et al., 2024).Additionally, we considered how 4- and 8-bit quantizations influence length representations in LLMs.

To investigate how explicit fine-tuning with length constraint prompts affects length-related internal representations within LLMs, we fine-tune a model on the Google sentence summarization dataset. Following previous work, which includes length-constraint prompts to enhance model's ability to control the output sequence length (Juseon-Do et al., 2024), we utilized QLoRA, a technique that can maintain the full 16-bit fine-tuning performance (Dettmers et al., 2023), for fine-tuning. QLoRA extends the Low-Rank Adapters (LoRA) method (Hu et al., 2022), which is an advanced form of parameter-efficient fine-tuning (Mangrulkar et al., 2022) for LLMs. This approach integrates low-rank, trainable matrices with the frozen weights of each transformer layer. During training, we used 8-bit quantization for QLoRA, while during inference, we employed 4-bit quantization. We used greedy decoding across all settings in our experiments to eliminate randomness in the generation process. Appendix A provides further details of hyper-parameters and settings.

---

[3]https://github.com/google-research-datasets/sentence-compression.git
[4]https://github.com/JuseonDo/InstructCMP

**Data Preparation.** An input sentence $\mathbf{S} = \{s_1, s_2, \ldots, s_n\}$ was first converted into vector embeddings, after which learned positional embeddings were added to form $\mathbf{S_{emb}} = \{s_1^e, s_2^e, \ldots, s_n^e\}$. These embeddings were then normalized using layer normalization, expressed as $S_{\text{norm}} = \text{LayerNorm}(\mathbf{S_{emb}})$. Then, they were computed through query ($\mathbf{W_Q}$), key ($\mathbf{W_K}$), and value ($\mathbf{W_V}$) matrices, and were fed into the transformer layers as follows:

$$\text{MultiHead}(Q, K, V) = \text{Concat}(\text{head}_1, \text{head}_2, \ldots, \text{head}_h)W_O \tag{1}$$

$$S_{\text{attn}} = \mathbf{S_{emb}} + \text{MultiHead}(S_{\text{norm}}\mathbf{W_Q}, S_{\text{norm}}\mathbf{W_K}, S_{\text{norm}}\mathbf{W_V}) \tag{2}$$

$$S_{\text{ffn}} = \text{ReLU}(\text{LayerNorm}(S_{\text{attn}})W_1 + b_1)W_2 + b_2 \tag{3}$$

$$S_{\text{out}} = S_{\text{attn}} + S_{\text{ffn}}, \tag{4}$$

where each $\text{head}_i = \text{softmax}\left(\frac{QK^T}{\sqrt{d_k}}\right)V$ indicates a self-attention operation.

We considered four outputs from the transformer layers, wherein each output represents a distinct level of encoded information derived from the original input sentence $\mathbf{S}$. We conducted sentence summarization using prompts in three different settings: No-constraint, Length, and Priming. For each setting, we investigated these four outputs for each layer. During token generation, we saved each output with its corresponding numeric time step value, excluding the input token prompts. For instance, we saved $n$ with its corresponding output when the model generated the $n$-th token. Appendix B provides further details for predicting time steps from hidden states.

The outputs from the multi-head attention in Equation (1) calculate attention scores between tokens, thus enabling the model to capture long-range dependencies. In Equation (2), the multi-head attention outputs are summed with the original embeddings. The outputs from Equation (3) use a feed-forward network (FFN) with a ReLU activation function. The final outputs from Equation (4) integrate the attention and FFN outputs.

**Neural Network Regression.** To find evidence of length representations in LLMs, we applied a standard technique to predict a target label associated with labeled input data (Shi et al., 2016), specifically, $\mathbf{X} \in \mathbb{R}^{n \times d_{\text{model}}}$, where $n$ refers to the number of data, $d_{\text{model}}$ is the dimensionality of a model's hidden states, and $\mathbf{Y}$ is a target that contains the generation time step as a numeric value for each corresponding $\mathbf{X}$. We used a two-layer neural network with a hidden layer of 100 neurons to predict $\hat{\mathbf{Y}} = \mathbf{W}_2(\text{ReLU}(\mathbf{W}_1\mathbf{X} + \mathbf{b}_1)) + \mathbf{b}_2$. By investigating how well the model can predict the generation time step, we can gain insights into how length representations are encoded within the LLM's hidden states. To assess how well the time step can be predicted from its corresponding hidden state in LLMs, we considered the coefficient determination, $R^2$, as a standard regression metric to evaluate the overall performance.

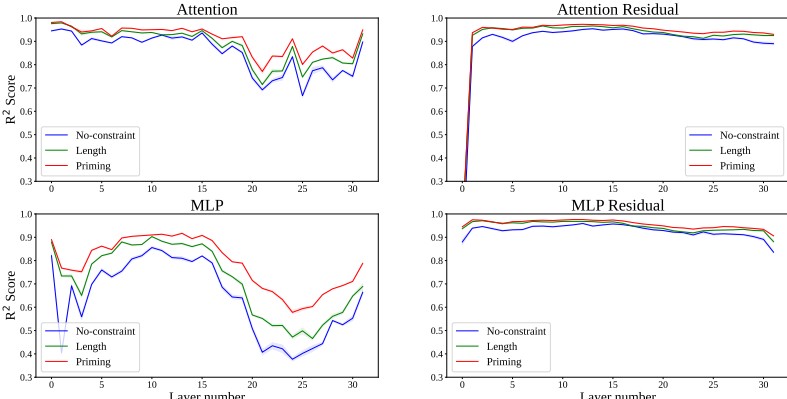

Figure 1: $R^2$ scores and their standard deviations based on five runs for outputs of four different types of transformer layers. The scores were averaged. Attention and attention residual refer to the outputs of Equations (1) and (2), respectively. MLP and MLP residual indicate the outputs of Equations (3) and (4), respectively.

Table 2: $R^2$ scores and their standard deviations based on five runs for different models with constraints. F and S indicate the first and second layers in LLMs, respectively. L indicates the last layer. Constraint indicates prompt types. The scores were averaged.

| Model | Constraint | Layer type | | | | | | | | | | | |
|---|---|---|---|---|---|---|---|---|---|---|---|---|---|
| | | Attn Out | | | Attn Residual | | | MLP Out | | | MLP Residual | | |
| | | F | S | L | F | S | L | F | S | L | F | S | L |
| Llama-2-7B | No-constraint | 0.94 (0.00) | **0.95** (0.00) | 0.88 (0.00) | 0.00 (0.00) | 0.90 (0.00) | 0.89 (0.00) | 0.84 (0.00) | 0.70 (0.00) | 0.67 (0.01) | 0.90 (0.00) | 0.94 (0.00) | 0.85 (0.00) |
| | Length | 0.98 (0.00) | **0.99** (0.00) | 0.93 (0.00) | 0.11 (0.00) | 0.94 (0.00) | 0.93 (0.00) | 0.89 (0.00) | 0.77 (0.00) | 0.70 (0.01) | 0.95 (0.00) | 0.97 (0.00) | 0.89 (0.00) |
| | Priming | 0.98 (0.00) | **0.99** (0.00) | 0.95 (0.00) | 0.11 (0.00) | 0.94 (0.00) | 0.94 (0.00) | 0.89 (0.00) | 0.77 (0.00) | 0.78 (0.00) | 0.95 (0.00) | 0.98 (0.00) | 0.92 (0.00) |
| Llama-2-13B | No-constraint | 0.95 (0.00) | **0.96** (0.00) | 0.93 (0.00) | 0.08 (0.01) | 0.93 (0.00) | 0.92 (0.00) | 0.90 (0.00) | 0.83 (0.00) | 0.74 (0.01) | 0.93 (0.00) | 0.95 (0.00) | 0.89 (0.00) |
| | Length | **0.94** (0.00) | **0.94** (0.00) | 0.92 (0.00) | 0.10 (0.00) | 0.92 (0.00) | 0.92 (0.00) | 0.89 (0.00) | 0.81 (0.00) | 0.75 (0.00) | 0.91 (0.00) | **0.94** (0.00) | 0.91 (0.00) |
| | Priming | **0.99** (0.00) | **0.99** (0.00) | 0.91 (0.00) | 0.17 (0.00) | 0.96 (0.00) | 0.92 (0.00) | 0.92 (0.00) | 0.81 (0.00) | 0.72 (0.00) | 0.97 (0.00) | 0.98 (0.00) | 0.89 (0.00) |
| Llama-2-13B (fine-tuned) | No-constraint | **0.99** (0.00) | **0.99** (0.00) | 0.88 (0.00) | 0.17 (0.01) | 0.97 (0.00) | 0.92 (0.00) | 0.93 (0.00) | 0.81 (0.00) | 0.74 (0.00) | 0.98 (0.00) | 0.98 (0.00) | 0.91 (0.00) |
| | Length | **0.99** (0.00) | **0.99** (0.00) | 0.87 (0.00) | 0.21 (0.01) | 0.97 (0.00) | 0.93 (0.00) | 0.92 (0.00) | 0.83 (0.00) | 0.78 (0.01) | 0.98 (0.00) | 0.98 (0.00) | 0.91 (0.00) |
| | Priming | **0.99** (0.00) | **0.99** (0.00) | 0.90 (0.01) | 0.16 (0.01) | 0.96 (0.00) | 0.93 (0.00) | 0.92 (0.00) | 0.85 (0.00) | 0.83 (0.00) | 0.97 (0.00) | 0.98 (0.00) | 0.92 (0.00) |
| Llama-2-70B | No-constraint | 0.97 (0.00) | **0.99** (0.00) | 0.95 (0.00) | 0.16 (0.00) | 0.93 (0.00) | 0.92 (0.00) | 0.83 (0.01) | 0.81 (0.01) | 0.82 (0.00) | 0.95 (0.00) | 0.98 (0.00) | 0.92 (0.00) |
| | Length | 0.97 (0.00) | **0.99** (0.00) | 0.94 (0.00) | 0.17 (0.00) | 0.92 (0.00) | 0.93 (0.00) | 0.87 (0.00) | 0.84 (0.00) | 0.80 (0.01) | 0.95 (0.00) | 0.98 (0.00) | 0.92 (0.00) |
| | Priming | **0.98** (0.00) | 0.97 (0.00) | 0.91 (0.00) | 0.18 (0.00) | 0.91 (0.00) | 0.89 (0.00) | 0.82 (0.00) | 0.76 (0.01) | 0.78 (0.01) | 0.94 (0.00) | 0.95 (0.00) | 0.88 (0.00) |
| Llama-3-8B | No-constraint | 0.96 (0.00) | **0.98** (0.00) | 0.91 (0.00) | 0.20 (0.00) | 0.86 (0.00) | 0.91 (0.00) | 0.70 (0.00) | 0.74 (0.00) | 0.78 (0.01) | 0.88 (0.00) | 0.95 (0.00) | 0.88 (0.00) |
| | Length | 0.96 (0.00) | **0.97** (0.00) | 0.93 (0.00) | 0.16 (0.00) | 0.88 (0.00) | 0.93 (0.00) | 0.72 (0.00) | 0.75 (0.00) | 0.79 (0.01) | 0.90 (0.00) | 0.96 (0.00) | 0.89 (0.00) |
| | Priming | 0.97 (0.00) | **0.98** (0.00) | 0.94 (0.00) | 0.24 (0.00) | 0.87 (0.00) | 0.94 (0.00) | 0.73 (0.00) | 0.76 (0.00) | 0.87 (0.00) | 0.89 (0.00) | 0.95 (0.00) | 0.92 (0.00) |
| Phi3-mini-4k | No-constraint | 0.93 (0.00) | **0.97** (0.00) | 0.91 (0.00) | 0.07 (0.01) | 0.80 (0.01) | 0.91 (0.00) | 0.61 (0.01) | 0.66 (0.01) | 0.55 (0.01) | 0.84 (0.00) | 0.95 (0.00) | 0.86 (0.00) |
| | Length | 0.94 (0.00) | **0.97** (0.00) | 0.92 (0.00) | 0.04 (0.00) | 0.80 (0.00) | 0.92 (0.00) | 0.65 (0.01) | 0.67 (0.00) | 0.56 (0.01) | 0.82 (0.00) | 0.94 (0.00) | 0.86 (0.00) |
| | Priming | 0.93 (0.00) | **0.97** (0.00) | 0.89 (0.00) | 0.07 (0.01) | 0.77 (0.00) | 0.90 (0.00) | 0.48 (0.01) | 0.63 (0.01) | 0.58 (0.01) | 0.80 (0.00) | 0.95 (0.00) | 0.84 (0.00) |
| Phi3-small-8k | No-constraint | 0.92 (0.01) | **0.95** (0.00) | 0.71 (0.01) | 0.01 (0.01) | 0.83 (0.00) | 0.83 (0.00) | 0.76 (0.01) | 0.72 (0.01) | 0.48 (0.00) | 0.87 (0.00) | 0.90 (0.00) | 0.81 (0.01) |
| | Length | 0.94 (0.00) | **0.97** (0.00) | 0.85 (0.00) | 0.11 (0.01) | 0.86 (0.00) | 0.87 (0.00) | 0.80 (0.00) | 0.79 (0.01) | 0.54 (0.01) | 0.89 (0.00) | 0.94 (0.00) | 0.87 (0.00) |
| | Priming | 0.97 (0.00) | **0.98** (0.00) | 0.81 (0.01) | 0.27 (0.01) | 0.92 (0.00) | 0.87 (0.01) | 0.86 (0.01) | 0.83 (0.00) | 0.58 (0.00) | 0.93 (0.00) | 0.97 (0.00) | 0.86 (0.00) |

## 4 LENGTH REPRESENTATIONS IN LLMs

We first investigated the following empirical questions: Which transformer layer in LLMs contains length information? Which outputs from a transformer layer in LLMs corresponding to Equations (1), (2), (3), and (4) contain length information? Do length-specific prompts influence length representations in LLMs? Do LLMs retain length representations when 4- and 8-bit quantizations are applied? How do these compare to full-precision models? Does fine-tuning influence length representations in LLMs?

### 4.1 LAYER-WISE ANALYSIS FOR LENGTH REPRESENTATIONS

Figure 1 shows the variation of $R^2$ for outputs from a transformer layer corresponding to Equations (1), (2), (3), and (4) using Llama-2-7B-Chat. It reveals an interesting pattern in the length representations within LLMs. In particular, in the second layer, the outputs of Equation (1), which indicates the attention mechanism, show a stronger correlation with the length representations than the outputs from Equations (2), (3), and (4). While the outputs from the other equations also provide strong correlations, the outputs from Equation (1) have consistently higher $R^2$ scores for all prompts. We also observed a gradual decrease in length representations as inputs are fed into subsequent LLM layers, and tokens are produced in a step-by-step process. However, in the final layer, the length representations begin to increase from the outputs of Equation (1). This result indicates that the LLM captures length representations in the early stages, similar to how they capture semantic representations (Niu et al., 2022). As such, the increase in length representations in the final layer indicates that the model may revisit this information to reinforce positional context.

### 4.2 INFLUENCE OF LENGTH-SPECIFIC PROMPTS

Table 2 shows the results of $R^2$ for outputs, which include Llama and Phi LLMs with a 4-bit quantization setting. The results reveal that the attention output consistently has higher $R^2$ scores than the other outputs, particularly in the second layer for the Llama- and Phi-3 families, regardless of model sizes. However, we observed a notable decrease in performance in the first layer, particularly in the attention residual. This indicates that the initial input sequence embeddings do not effectively contain length representations; however, these representations progressively accumulate through the layers. Although the length-specific prompting method (Priming) can precisely control output sequence length (Juseon-Do et al., 2024), it does not increase the $R^2$ when using all hidden units for prediction. However, when we fine-tuned the models, we found that every model, regardless of the prompts used, the $R^2$ scores were improved.

Table 3: $R^2$ scores with 8-bit and full-precision settings. In each cell, x/y represents the 8-bit quantization and full-precision. The notations are the same as those in Table 2 and standard deviations are nearly zero.

| Model | Constraint | Attn Out | | | Attn Residual | | | MLP Out | | | MLP Residual | | |
|---|---|---|---|---|---|---|---|---|---|---|---|---|---|
| | | F | S | L | F | S | L | F | S | L | F | S | L |
| Llama-2 7B | No-constraint | **0.99**/0.98 | **0.99**/**0.99** | 0.93/0.92 | 0.11/0.09 | 0.95/0.94 | 0.94/0.93 | 0.89/0.90 | 0.73/0.73 | 0.72/0.73 | 0.95/0.95 | 0.98/0.98 | 0.89/0.89 |
| | Length | 0.98/**0.99** | **0.99**/**0.99** | 0.94/0.94 | 0.11/0.10 | 0.94/0.94 | 0.94/0.93 | 0.89/0.89 | 0.75/0.76 | 0.72/0.73 | 0.95/0.95 | 0.98/0.98 | 0.90/0.90 |
| | Priming | 0.98/0.98 | **0.99**/**0.99** | 0.94/0.95 | 0.14/0.13 | 0.94/0.94 | 0.94/0.94 | 0.90/0.91 | 0.77/0.77 | 0.77/0.77 | 0.95/0.96 | 0.98/0.98 | 0.91/0.91 |
| Llama-2 13B | No-constraint | 0.58/0.55 | 0.70/0.68 | **0.76**/**0.74** | 0.01/0.04 | 0.58/0.56 | 0.73/0.72 | 0.59/0.58 | 0.56/0.57 | 0.61/0.59 | 0.58/0.55 | 0.66/0.64 | 0.70/0.69 |
| | Length | **0.99**/**0.99** | **0.99**/**0.99** | 0.94/0.94 | 0.11/0.11 | 0.96/0.97 | 0.94/0.94 | 0.92/0.93 | 0.83/0.83 | 0.76/0.76 | 0.96/0.96 | 0.98/0.98 | 0.92/0.92 |
| | Priming | **0.99**/**0.99** | **0.99**/**0.99** | 0.92/0.92 | 0.19/0.19 | 0.96/0.96 | 0.92/0.91 | 0.92/0.91 | 0.80/0.81 | 0.75/0.76 | 0.96/0.97 | 0.98/0.98 | 0.90/0.89 |
| Llama-2 (13B-tuned) | No-constraint | **0.99**/**0.99** | 0.98/0.98 | 0.87/0.87 | 0.19/0.22 | 0.96/0.97 | 0.92/0.92 | 0.91/0.92 | 0.80/0.80 | 0.75/0.77 | 0.97/0.97 | 0.99/0.98 | 0.90/0.90 |
| | Length | **0.99**/**0.99** | 0.98/0.99 | 0.87/0.87 | 0.19/0.20 | 0.96/0.97 | 0.92/0.93 | 0.91/0.91 | 0.81/0.83 | 0.78/0.78 | 0.97/0.97 | 0.98/0.98 | 0.91/0.92 |
| | Priming | **0.99**/**0.99** | **0.99**/0.98 | 0.90/0.90 | 0.19/0.19 | 0.95/0.96 | 0.93/0.93 | 0.91/0.92 | 0.86/0.86 | 0.82/0.82 | 0.96/0.97 | 0.98/0.98 | 0.92/0.92 |
| Llama-3 8B | No-constraint | 0.96/0.93 | **0.97**/0.96 | 0.92/0.92 | 0.18/0.18 | 0.86/0.82 | 0.91/0.91 | 0.69/0.69 | 0.76/0.75 | 0.79/0.80 | 0.87/0.83 | 0.95/0.93 | 0.88/0.88 |
| | Length | 0.96/0.95 | **0.98**/**0.97** | 0.93/0.93 | 0.15/0.15 | 0.87/0.86 | 0.92/0.93 | 0.70/0.72 | 0.78/0.76 | 0.78/0.79 | 0.89/0.87 | 0.96/0.95 | 0.89/0.89 |
| | Priming | 0.88/0.86 | **0.91**/0.85 | 0.90/**0.86** | 0.16/0.14 | 0.79/0.64 | 0.84/0.82 | 0.64/0.53 | 0.60/0.55 | 0.59/0.61 | 0.79/0.70 | 0.87/0.79 | 0.75/0.73 |
| Phi3 mini-4k | No-constraint | 0.94/0.94 | **0.97**/**0.97** | 0.92/0.92 | 0.07/0.07 | 0.80/0.82 | 0.93/0.93 | 0.61/0.64 | 0.66/0.67 | 0.52/0.53 | 0.84/0.84 | 0.95/0.96 | 0.87/0.87 |
| | Length | 0.94/0.94 | **0.97**/**0.97** | 0.93/0.93 | 0.06/0.05 | 0.82/0.82 | 0.93/0.92 | 0.61/0.63 | 0.67/0.65 | 0.53/0.52 | 0.84/0.83 | 0.95/0.96 | 0.87/0.86 |
| | Priming | 0.92/0.93 | **0.97**/**0.98** | 0.90/0.90 | 0.10/0.12 | 0.74/0.75 | 0.90/0.90 | 0.48/0.46 | 0.61/0.66 | 0.58/0.60 | 0.78/0.78 | 0.94/0.96 | 0.84/0.85 |
| Phi3 small-8k | No-constraint | 0.95/0.95 | **0.96**/**0.96** | 0.74/0.73 | 0.03/0.06 | 0.88/0.89 | 0.84/0.86 | 0.80/0.79 | 0.76/0.74 | 0.45/0.47 | 0.90/0.91 | 0.94/0.94 | 0.84/0.85 |
| | Length | 0.95/0.97 | **0.96**/0.97 | 0.84/0.86 | 0.10/0.12 | 0.87/0.90 | 0.88/0.90 | 0.80/0.82 | 0.77/0.80 | 0.55/0.56 | 0.89/0.92 | 0.94/0.96 | 0.87/0.90 |
| | Priming | 0.97/**0.97** | **0.98**/0.97 | 0.81/0.82 | 0.29/0.31 | 0.92/0.92 | 0.87/0.89 | 0.86/0.86 | 0.83/0.82 | 0.58/0.59 | 0.92/0.94 | 0.96/**0.97** | 0.87/0.89 |

## 4.3 QUANTIZATION ON LENGTH REPRESENTATIONS

Table 3 shows the $R^2$ results for outputs with 8-bit and full-precision settings. We observed that the results are similar to those obtained with 4-bit quantization, wherein length representations are more prominently encoded in the attention outputs from the second layer than the other outputs. This indicates that whether 4- or 8-bit quantization is applied does not significantly affect the LLMs' capabilities to encode length representations. Therefore, the attention mechanism of the second layer consistently captures length representations across different precision levels even for different models with varying sizes.

## 5 EFFECT OF DISENTANGLING LENGTH REPRESENTATIONS

The previous section investigated which components and layers contain length representations for output sequence length with varying prompts. While we found that the second layer in the attention networks has a strong correlation with length representations, this does not indicate which hidden units are actually responsible for controlling the output sequence length. Thus, which hidden units must be identified for a better understanding of LLMs' length control.

### 5.1 DO LENGTH-SPECIFIC PROMPTS AFFECT INNER LENGTH REPRESENTATIONS?

We also trained separate regressions on each single hidden unit from the second layer of the attention outputs in Llama-2-13B-Chat, which has a total of 5,120 hidden units. Table 4 shows the results of the top-10 highest $R^2$ scores based on prediction from individual hidden units to each time step. Unlike all units that were used for prediction in Table 2, we observed distinct differences among the constraints, including the length-specific prompting method (Priming).

Compared to the No-constraint and Length prompting methods, length-related hidden units within the attention networks become more active in representing length information when we used more length-specific prompts such as Priming. Moreover, in the zero-shot setting, the No-constraint and Length prompts share nearly the same top-10 hidden units for representing length information. Conversely, when the Priming prompt was used, different hidden units become active, thus indicating a shift in how length information is captured by the LLM with more length-specific prompting. Additionally, more length-specific prompts of Priming cause each of the top-$k$ hidden units to be more highly activated than those in the No-constraint and Length prompt settings.

Furthermore, the hidden units for representing length information became nearly identical across prompting methods when we fine-tuned LLMs, because the model learned the capability to precisely control output sequence length. Interestingly, the same top-3 hidden units are activated with the Priming prompt in the zero-shot and fine-tuning settings. This finding indicates that specific length-related units are consistently activated during Priming, thus guiding LLMs to manage the generation of output sequence length. These empirical results further demonstrate that LLMs understand in-context learning as a form of implicit fine-tuning (Dai et al., 2023).

Table 4: $R^2$ scores for individual hidden units from the Llama-2-13B-Chat with a 4-bit quantization setting. The numbers in parentheses indicate an index of hidden units from the second layer of the attention mechanisms.

| Setting | Prompting | $1^{st}$ | $2^{nd}$ | $3^{rd}$ | $4^{th}$ | $5^{th}$ | $6^{th}$ | $7^{th}$ | $8^{th}$ | $9^{th}$ | $10^{th}$ | Avg 30 |
|---|---|---|---|---|---|---|---|---|---|---|---|---|
| | No-constraint | 0.11 (2,100) | 0.10 (110) | 0.09 (435) | 0.07 (3,499) | 0.07 (190) | 0.07 (1,160) | 0.06 (3,459) | 0.06 (1,611) | 0.06 (4,775) | 0.06 (4,305) | 0.06 |
| Zero-shot | Length | 0.14 (2,100) | 0.10 (110) | 0.06 (435) | 0.05 (321) | 0.05 (1,411) | 0.05 (1,611) | 0.05 (4,775) | 0.05 (3,499) | 0.04 (1,069) | 0.04 (4,832) | 0.05 |
| | Priming | **0.38** (371) | **0.32** (2,741) | **0.23** (1,380) | **0.19** (4,698) | **0.18** (4,554) | **0.15** (1,728) | **0.13** (4,923) | **0.13** (2,100) | **0.09** (2,846) | **0.08** (5,046) | **0.08** |
| | No-constraint | **0.42** (2,741) | 0.35 (1,380) | **0.34** (371) | 0.28 (4,698) | **0.26** (2,282) | **0.25** (4,372) | **0.25** (1,419) | 0.23 (614) | **0.22** (5,046) | 0.21 (1,728) | 0.19 |
| Fine-tuning | Length | 0.39 (1,380) | 0.38 (371) | 0.37 (2,741) | 0.28 (4,372) | 0.25 (4,698) | 0.24 (614) | 0.21 (2,282) | 0.21 (1,419) | 0.21 (1,728) | 0.20 (31) | 0.20 |
| | Priming | 0.40 (371) | **0.39** (2,741) | **0.34** (1,380) | **0.31** (4,372) | **0.26** (1,419) | **0.25** (614) | 0.24 (1,030) | **0.23** (1,728) | 0.21 (5,046) | **0.21** (2,282) | **0.21** |

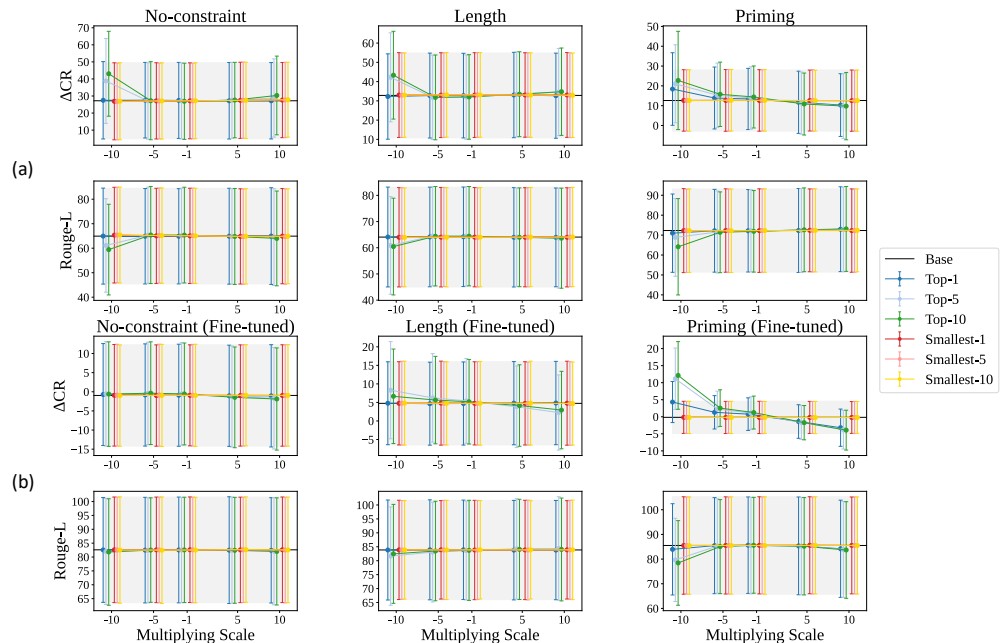

Figure 2: ΔCR and Rouge-L scores change with standard deviations by multiplying scale in the Llama-2-13B-Chat. (a) and (b) mean zero-shot and fine-tuning settings, respectively. Base means original scores without scale modification (i.e., the multiplying scale is 1). The color gray represents the standard deviations of the Base.

## 5.2 DOES SCALING LENGTH REPRESENTATIONS AFFECT MODEL-GENERATED TEXT?

To investigate the effect of scaling length representations on model-generated text, we disentangled the top-$k$ and smallest-$k$ activated hidden units related to length representations in the second layer from multi-head attention mechanisms. We multiplied these hidden units by negative or positive numeric values to adjust the output sequence length. This scaling was applied to all output token positions during the generation process, excluding the input token prompts. We aimed to empirically demonstrate that such hidden units consider length representations in LLMs and are partially disentangled from semantic representations. We used Rouge-L (R-L) (Lin, 2004) to evaluate the informativeness of the summarized sentences when we scaled length representations. R-L measures the longest common subsequence between the generated and gold summaries. To evaluate the performance with respect to length, we used ΔCR, which is an arithmetical difference between the model-generated and gold compression ratios. Thus, ΔCR value close to zero indicates that the model-generated summary has a compression ratio similar to that of the gold summary. A higher ΔCR means the generated summary is longer than the gold summary, while a lower ΔCR means it is shorter.

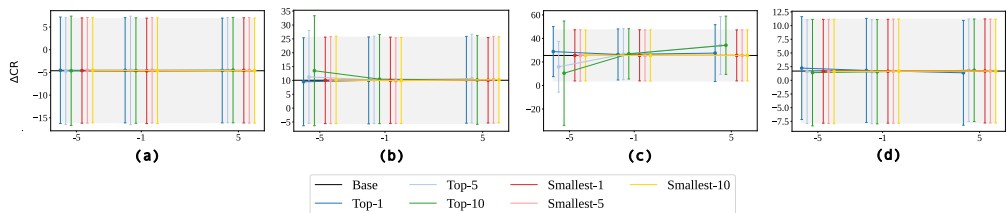

Figure 3: Results using different LLMs in zero-shot settings. Models used are (a) Phi-3 Small-8k, (b) Phi-3 Mini-4k, (c) Llama-2-7B, and (d) Llama-3-8B.

Figure 2 (a) presents the results of applying negative or positive scaling factors to the top-$k$ and smallest-$k$ activated hidden units in zero-shot settings. When using more length-specific prompts of Priming, we observed more consistent changes of $\Delta$CR with modifying only the top-1 hidden unit. We think this is because Priming contains more highly activated hidden units related to length representations than No-constraint and Length. Thus, the highly activated length-related units provide better length representations and length controllability, as shown in Table 4. Additionally, the output sequence length changes according to increases in the scaling factor. While multiplying positive and negative values enables the LLM to produce shorter and longer summaries, respectively, than the original hidden units, particularly in the Priming prompt, in the No-constraint and Length prompts, the LLM does not generate shorter summaries even when positive scaling values were applied. As for R-L scores, disentangling the top-$k$ units improves performance, particularly in the Priming prompt. This finding indicates that adjusting the most highly activated length-related units not only controls length but also enhances the informativeness of the generated text. However, when we applied large scaling factors, such as -10 or 10, the R-L scores slightly decrease when the No-constraint and Length prompts were used. In comparison, for Priming, which is more length-specific prompts, continues to improve performance even when we applied a large scaling factor of 10. Finally, disentangling the smallest-$k$ units does not lead to significant changes in output sequence length, thus indicating that these units are less involved in encoding length information. For selected smallest-$k$ units, the individual $R^2$ scores are nearly 0.

Figure 2 (b) shows the results of applying negative or positive scaling factors to the top-$k$ and smallest-$k$ activated hidden units in fine-tuned settings. In contrast to the previous zero-shot settings, we obtained more stable results for all prompts when we disentangled the hidden units. While multiplying positive scaling values results in generating shorter summaries, multiplying negative scaling values produces longer summaries. This is because fine-tuning has strengthened the LLM's reliance on the top-$k$ length-related units for precise length control. We notice that scaling with large factors caused a slight decrease in R-L scores, particularly in the No-constraint prompt. While large scaling factors lead to greater changes in $\Delta$CR and R-L for the Priming prompt, higher overall R-L scores are maintained. Furthermore, disentangling the smallest-$k$ has minimal impact on sequence length among all prompts. Specifically, there are no significant changes in output sequence length when the smallest-$k$ hidden units were modified. Appendix C provides further details of the top-$k$ and smallest-$k$.

Figure 3 shows the results using different LLMs in zero-shot settings. When we scale the top-$k$ hidden units by multiplying them with scaling factors, we observe variations in output length. In contrast, scaling the smallest-$k$ hidden units does not impact length control during generation.

## 5.3 BIN-WISE ANALYSIS OF EXTREME SCALING FACTORS ON LENGTH VARIATION

To investigate the effect of applying extreme scaling factors to the top-$k$ hidden units, we analyzed the results by grouping them into bins with the word count of summaries, which were generated using a white-space-based split and the base scale. Table 5 shows the results. Multiplying positive values enables the LLM to generate shorter summaries, while multiplying negative values results in longer summaries, especially with the Priming prompts in the fine-tuning settings. Furthermore, using the Base maintains longer summaries, and the greater length variations were observed when we multiplied scales. This explains why the zero-shot setting shows greater variations in $\Delta$CR scores, as shown in Figures 2 and 5, even though each hidden unit has a lower $R^2$ score than the fine-tuning setting. Scaling with extreme factors to the top-$k$ often results in decreased informative-

Table 5: Results based on word length in Llama-2-13B-Chat with the Priming prompts. In each cell, x/y represents $\Delta$CR and R-L. "#Data" refers to the number of data in that interval. [†] indicates the improvement is significant (p<0.05) compared with the Base when generating shorter or longer outputs with positive or negative factors, respectively. (Koehn, 2004).

| Setting | Prompting | Scale | # of words generated by the Base scale | | |
| | | | 1-10 | 11-20 | 21- |
| | | #Data | 154 | 559 | 287 |
| Zero-shot | No-constraint | Base | 1.28 (17.49) / 74.61 (22.58) | 23.92 (17.30) / 68.99 (17.25) | 47.48 (13.86) / 51.83 (14.74) |
| | | 10 | 8.15 (21.27) / 72.93 (21.92) | 27.56 (19.42) / 67.77 (17.14) | 47.60 (17.32) / 51.83 (15.94) |
| | | -10 | 25.27[†] (27.64) / 68.13 (20.34) | 40.61[†] (20.98) / 62.47 (17.09) | 57.32[†] (22.50) / 48.83 (15.24) |
| | | #Data | 93 | 557 | 350 |
| | Length | Base | 4.22 (16.10) / 77.69 (23.42) | 26.50 (17.86) / 69.53 (16.51) | 50.49 (14.72) / 51.77 (14.42) |
| | | 10 | 9.83 (20.39) / 75.97 (22.99) | 28.56 (19.43) / 69.08 (16.81) | 51.26 (16.17) / 51.74 (14.97) |
| | | -10 | 19.85[†] (24.09) / 72.95 (21.98) | 38.40[†] (20.83) / 64.97 (16.80) | 57.42[†] (16.00) / 49.92 (14.57) |
| | | #Data | 357 | 558 | 85 |
| | Priming | Base | 2.99 (9.78) / 74.79 (24.45) | 15.04 (13.34) / 72.56 (18.33) | 37.14 (14.07) / 60.14 (14.80) |
| | | 10 | 1.51[†] (10.27) / 75.33 (23.82) | 12.22[†] (16.71) / 73.15 (19.36) | 28.43[†] (21.24) / 63.49[†] (19.65) |
| | | -10 | 12.72[†] (22.26) / 65.85 (27.94) | 26.18[†] (24.21) / 63.96 (22.34) | 42.20[†] (20.71) / 58.15 (16.60) |
| | | #Data | 649 | 341 | 10 |
| Fine-tuning | No-constraint | Base | -4.38 (11.96) / 82.32 (19.75) | 4.57 (11.95) / 83.64 (17.33) | 32.61 (19.87) / 65.20 (17.21) |
| | | 10 | -4.75[†] (12.12) / 82.02 (19.63) | 2.66[†] (12.82) / 82.66 (18.04) | 28.20 (23.41) / 60.28 (26.72) |
| | | -10 | -3.14[†] (13.11) / 81.75 (19.72) | 3.50 (12.62) / 82.52 (17.58) | 23.34 (22.14) / 61.13 (24.83) |
| | | #Data | 492 | 499 | 9 |
| | Length | Base | 1.14 (9.30) / 84.74 (19.03) | 7.96 (11.60) / 83.09 (16.46) | 24.02 (11.73) / 78.07 (18.05) |
| | | 10 | 1.09 (9.11) / 84.86 (18.96) | 4.57[†] (11.28) / 83.32 (17.79) | 14.03[†] (9.41) / 77.85 (22.60) |
| | | -10 | 3.49[†] (11.71) / 82.69 (18.93) | 9.46[†] (12.82) / 82.30 (16.51) | 23.45 (13.64) / 79.22 (17.62) |
| | | #Data | 629 | 370 | 1 |
| | Priming | Base | -0.36 (4.08) / 86.53 (20.55) | 0.18 (5.63) / 83.82 (18.21) | 0.00 (0.00) / 83.72 (0.00) |
| | | 10 | -2.92[†] (5.01) / 85.13 (20.52) | -5.61[†] (6.73) / 81.25 (17.64) | -8.70[†] (0.00) / 78.05 (0.00) |
| | | -10 | 9.12[†] (8.72) / 78.87 (19.06) | 17.24[†] (9.75) / 77.70 (13.28) | 15.22[†] (0.00) / 84.00[†] (0.00) |

ness, which indicates that length representations in LLMs are partially disentangled from semantic representations.

## 5.4 HUMAN EVALUATION AND CASE STUDY

We further investigated the robustness of scaling top-*k* approach by evaluating actually generated outputs with human evaluations.

**Human Evaluation Settings.** We conducted human evaluations to further assess the effect of disentangling length-related units in zero-shot and fine-tuning settings. Note that we separately evaluated the zero-shot and fine-tuning settings; thus, their scales might be different. We sampled 100 instances for each setting from the Google test dataset. Using Amazon Mechanical Turk, we assigned a total of 80 evaluators who held both US high school and bachelor's degrees for grading the results, with scores from 1 to 5 (5 is the best), in terms of Coherence (Coh), Conciseness (Conc), and informativeness (Info).

**Results.** In the zero-shot and fine-tuning settings, adjusting the length-related hidden units with positive scaling factors generally enhances conciseness but slightly decreases informativeness because of the process of generating shorter summaries by keeping coherence. In contrast, negative scaling improves informativeness but slightly decreases conciseness due to the production of longer summaries, which can be an inherent trade-off between conciseness and informativeness when controlling output sequence length in summarization (Kikuchi et al., 2016; Makino et al., 2019).

Table 6: The results of human evaluations using the Priming prompt. The notations are the same as those in Table 5 for scales between 10 and -10.

| Scale | Zero-shot | | | Fine-tuning | | |
| | Coh. | Conc. | Infor. | Coh. | Conc. | Infor. |
| -10 | 3.73 (0.25) | 3.56 (0.25) | **3.71**[†] (0.27) | 3.43 (0.25) | 3.33 (0.27) | **3.34**[†] (0.21) |
| 1 | 3.72 (0.29) | 3.59 (0.23) | 3.70 (0.29) | 3.48 (0.23) | 3.46 (0.20) | 3.31 (0.27) |
| Gold | 3.67 (0.30) | 3.58 (0.23) | 3.68 (0.27) | 3.42 (0.26) | 3.45 (0.24) | 3.28 (0.22) |
| 10 | 3.69 (0.24) | 3.59 (0.29) | 3.63 (0.29) | 3.41 (0.22) | **3.47**[†] (0.22) | 3.19 (0.23) |

**Case Study.** We conducted a detailed case study to analyze the effects of disentangling length-related hidden units by comparing the generated outputs for different scaling factors with the source and gold summaries. Figure 4 presents examples.

In the first example, we observed changes in the generated summaries based on different scaling factors. In particular, when negative scaling was applied, the generated summaries became longer than the Base summary by incorporating redundant information, such as "*Texans Coach Gary Kubiak said Thursday,*" from the source. In comparison, applying positive scaling values leads to shorter summaries by focusing on important content similar to the gold summary. When we disen-

tangled the smallest-$k$ hidden units, the generated summaries remained unchanged, regardless of the scaling factors, consistently producing the same summary as the Base. In the second example, the

| Type | | Text | Length (#word) |
|---|---|---|---|
| Source | | Case Keenum will start at quarterback Sunday for the Houston Texans in place of the injured Matt Schaub, Texans Coach Gary Kubiak said Thursday. | 24 |
| Gold | | Case Keenum will start at quarterback for the Houston Texans. | 10 |
| Top-10 | Scale 5 | Case Keenum will start at quarterback Sunday for the Texans in place of the injured Matt Schaub. | 14 (-3) |
| | Scale 10 | Case Keenum will start at quarterback Sunday for the Texans in place of the injured Matt Schaub, Kubiak said. | 15 (-2) |
| | Scale -5 | Case Keenum will start at quarterback Sunday for the Texans in place of the injured Matt Schaub, Texans Coach Gary Kubiak said Thursday. | 23 (+6) |
| | Scale -10 | Case Keenum will start at quarterback Sunday for the Houston Texans in place of the injured Matt Schaub, Texans Coach Gary Kubiak said Thursday. | 24 (+7) |
| Base (Scale 1) | | Case Keenum will start at quarterback Sunday for the Texans in place of the injured Matt Schaub. | 17 |
| Smallest -10 | Scale 5 | Case Keenum will start at quarterback Sunday for the Texans in place of the injured Matt Schaub. | 17 |
| | Scale 10 | Case Keenum will start at quarterback Sunday for the Texans in place of the injured Matt Schaub. | 17 |
| | Scale -5 | Case Keenum will start at quarterback Sunday for the Texans in place of the injured Matt Schaub. | 17 |
| | Scale -10 | Case Keenum will start at quarterback Sunday for the Texans in place of the injured Matt Schaub. | 17 |
| Source | | Armenian national's midfielder Aras Ozbiliz may miss the friendly match against Russia, technical director Vardan Minasyan told reporters ahead of the match. | 22 |
| Gold | | Aras Ozbiliz may miss the friendly match against Russia. | 9 |
| Top-10 | Scale 5 | Armenian midfielder may miss Russia against match. | 6 (-1) |
| | Scale 10 | Armenian midfielder may miss Russia against match. | 6 (-1) |
| | Scale -5 | Armenian midfielder Aras Ozbiliz may miss the match against Russia. | 10 (+3) |
| | Scale -10 | Aras Ozbiliz may miss the friendly match against Russia, technical director Vardan Minasyan told reporters ahead of the match. | 19 (+12) |
| Base (Scale 1) | | Armenian midfielder may miss match against Russia. | 7 |
| Smallest -10 | Scale 5 | Armenian midfielder may miss match against Russia. | 7 |
| | Scale 10 | Armenian midfielder may miss match against Russia. | 7 |
| | Scale -5 | Armenian midfielder may miss match against Russia. | 7 |
| | Scale -10 | Armenian midfielder may miss match against Russia. | 7 |

Figure 4: Summarization examples by scaling with Llama-2-13B-Chat in zero-shot Priming. The highlighted part represents the changed part from the Base text. The gray and red tokens indicate deleted and added tokens, respectively, while the blue token represents tokens that have changed their positions.

results are similar to the first example. However, when the Base summary is already short, positive scaling with a larger factor, such as 10, did not necessarily produce a shorter summary than a factor of 5. Furthermore, the LLM considers grammaticality when we applied scaling. For instance, when scaling with -5, the model generates "the" to maintain grammatical correctness. Appendix D provides other cases when multiplying extreme scales.

## 6 DISCUSSION AND CONCLUSION

In this study, we investigated the process by which output sequence length information is encoded within the internal representations of LLMs. We focused on identifying the specific components within transformer layers that contribute to length control during text generation tasks, particularly, sentence summarization. Our findings empirically demonstrated that the outputs from the second layer's attention mechanisms showed a strong correlation with the generation time step, thus indicating that length representations were captured early in the process. We also found that this pattern was consistent with different models with different sizes, such as the Llama and Phi families, and continued to be robust even when 4- and 8-bit quantizations were applied.

Furthermore, we analyzed individual hidden units from the second layer attention outputs and found that certain hidden units are highly activated and directly contributed to the process of representing length information. Moreover, these units became more active when length-specific prompts such as Priming were used. This finding indicates that LLMs adjust their internal representations based on the input prompts. Furthermore, by scaling these length-related hidden units, we effectively controlled the output sequence length without losing informativeness. While positive scaling factors led to shorter summaries, negative scaling resulted in longer summaries. It indicates that length information is partially disentangled from semantic representations within LLMs.

Finally, our results revealed that fine-tuning further improved the LLMs' capabilities by reinforcing reliance on the top-$k$ length-related units. We also found the same activation of specific hidden units in the Priming prompt are shared between zero-shot and fine-tuning settings, that indicates LLMs have constructed robust internal mechanisms for controlling output sequence length, and in-context learning performs similarly to implicit fine-tuning (Dai et al., 2023). Our findings have important implications for the interpretability and controllability of LLMs in natural language generation tasks. Understanding how length information is internally encoded allows for more precise length control over generated outputs, which is crucial in applications, such as summarization and machine translation, where adhering to length constraints is often required.

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

## A  EXPERIMENTAL DETAILS

Table 7: Hyperparameters

| Parameter | Value |
|---|---|
| Epochs | 1,000 |
| Batch size | 32, 64 |
| Learning rate | 1e-3 |
| Dropout rate | 0.1 |
| Patience | 5, 10 |
| Loss | MSE |
| Activation | ReLU |

**Computing interfaces.** We used the following GPUs:

- NVIDIA A100 GPU for Llama-2-70B-Chat
- NVIDIA A6000 GPU for other LLMs

**Hyperparameters.** Table 7 shows the hyperparameters used in our experiments. All feedforward networks were trained for 1,000 epochs with a learning rate of 1e-3 and a dropout rate of 0.1. For the linear regression to predict the generation time step from all hidden unit (Table 2, Table 3, and Figure 1), the batch size was set to 32 with an early stopping patience of 10 epochs. For the linear regression to predict the generation time step from each individual hidden unit (Table 4, Table 5, and Figure 2), the batch size was set to 64 with an early stopping patience of 5 epochs.

## B  DATASET DETAILS

Table 8: Dataset statistics

| Model | No-constraint | Length | Priming |
|---|---|---|---|
| Llama-2-7B | 21,385 | 24,121 | 25,394 |
| Llama-2-13B | 26,526 | 28,590 | 20,291 |
| Llama-2-13B(fine-tuned) | 14,826 | 16,373 | 14,884 |
| Llama-2-70B | 20,885 | 15,707 | 19,870 |
| Llama-3-8B | 17,952 | 22,853 | 13,366 |
| Phi3-mini-4k | 30,552 | 18,500 | 25,160 |
| Phi3-small-8k | 25,578 | 30,938 | 18,841 |

In our experiments, we randomly divided the datasets into 90% for training and 10% for validation. In the linear regression using all hidden unit, we used the entire dataset generated from each sequence of summaries. In contrast, when performing linear regression using individual hidden units, we restricted the number of dataset across different prompts to ensure that each model learned length information from the same amount of dataset.

## C  ADDITIONAL EXPERIMENTS

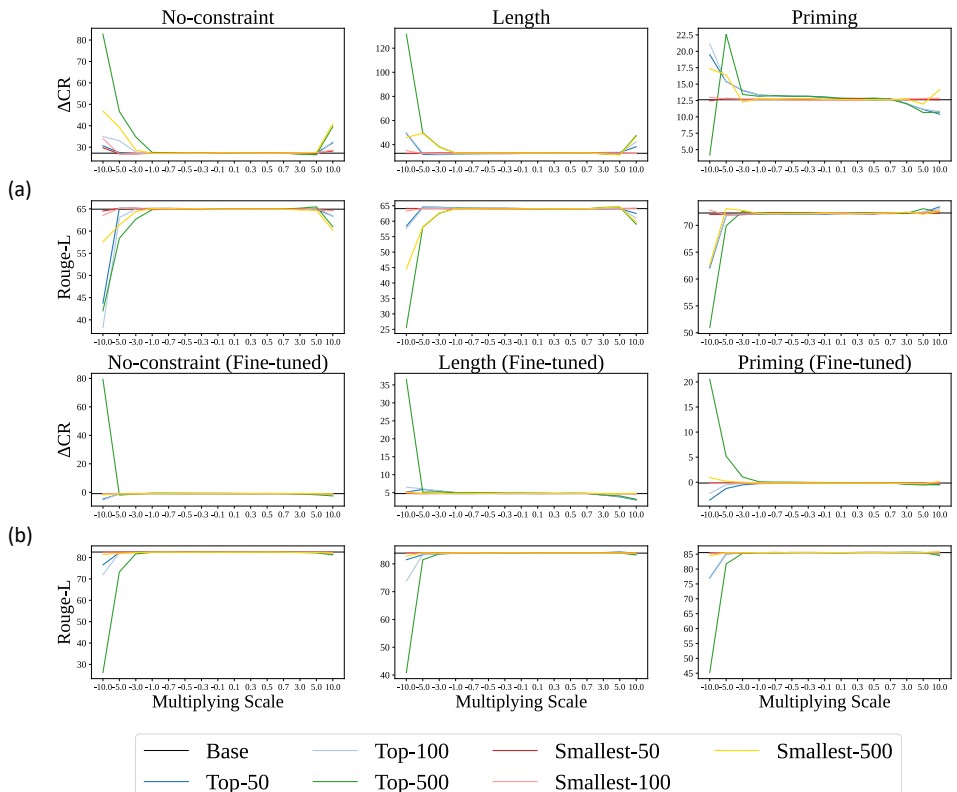

Figure 5: $\Delta$CR and Rouge-L score change by multiplying scale in Llama-2-13B-Chat, for (a) means zero-shot and (b) means fine-tuning setting. The notations are same as those in Figure 2

Table 9: Average $R^2$ scores of individual hidden unit for Top-500 with Llama-2-13B-Chat

| Setting | No-constraint | Length | Priming |
|---------|---------------|--------|---------|
| Zero-shot | 0.028 | 0.023 | 0.001 |
| Fine-tuning | 0.033 | 0.072 | 0.085 |

Figure 5 shows additional experimental results for disentangling the top- and smallest-50, -100, -500 hidden units. We obtained the similar results to that of Figure 2. When a large number of hidden units are modified, the top-$k$ still produce more significant length changes than the smallest-$k$. However, we also observed length changes when disentangling the smallest-$k$ units in zero-shot settings. This is because not only informativeness but also length information could be affected when many hidden units are modified, resulting in the significant decrease in R-L scores.

As shown in Table 9, when we averaged the top-500 hidden units of their individual $R^2$, we observed nearly zero $R^2$ scores for each setting; thus, disentangling top-500 hidden units resulted in significant decreases in R-L scores. While disentangling the top-500 hidden units led to large variations in output length, modifying the smallest-500 hidden units did not affect length variations in the fine-tuning settings.

## D  OTHER CASE STUDY

Figure 6 shows case studies. We found that the generated summaries ended abnormally early or that tokens were generated without spaces when extreme numeric values, such as -10, were used. This resulted in cases where the R-L scores significantly decreased with extreme scaling factors.

| Type | | Text | Length (#word) |
|---|---|---|---|
| Source | | South African captain Graeme Smith hailed ``an incredible win'' for his team after they clinched an emphatic ten-wicket victory on the fifth day of the second and final Test against India at Kingsmead on Monday. | 35 |
| Gold | | Graeme Smith hailed an incredible win. | 6 |
| Top-10 | Scale -10 | S. | 1 (-8) |
| Base (Scale 1) | | South African captain Graeme Smith hailed an incredible win. | 9 |
| | | | |
| Source | | Unknown assailants blew up a natural gas pipeline in Egypt, a security source said. | 14 |
| Gold | | Assailants blew up a gas pipeline in Egypt. | 8 |
| Top-10 | Scale -10 | AssBlewUpNatGasPipEgy. | 1 (-8) |
| Base (Scale 1) | | Assailants blew up a natural gas pipeline in Egypt. | 9 |

Figure 6: Case studies by scaling factors using Llama-2-13B-Chat with zero-shot priming.

