# OpenReview forum: "Length Representations in Large Language Models"
_ICLR.cc/2025/Conference — ICLR 2025 Conference Withdrawn Submission_

### Official Review · Reviewer_KeXg · 2024-10-31

**Soundness:** 3
**Presentation:** 2
**Contribution:** 2
**Rating:** 5
**Confidence:** 3

**Summary:**

This paper aims to understand how models encode and regulate the length of their generations, with the aim of controlling the length of model outputs. To do so, it first trains probes to predict the current generation length from model representations. Then, the authors identify individual neurons that are predictive of current generation length. By scaling these neurons' activations, they attempt to control generation length, while preserving the generation semantics and coherence. To verify that they have done so, they run a human evaluation study, and perform qualitative analysis of model generations under different levels of scaling.

**Strengths:**

This paper tackles a question—controlling the length of generations—that is important and, to the best of my knowledge, still open. It attempts to provide contributions in both interpretability and controllability, demonstrating potential for real-world impact. It does so using a variety of methods: probing, causal interventions, human studies, and some qualitative analysis of outputs too.

**Weaknesses:**

The most serious issues with this paper are presentation-related. Because the methods and precise results are unclear, it's difficult to judge the validity of the claims made in the paper. If the methods and results are made clearer, I will be able to revise my score. I'm putting the most important questions here, but see the others in **Questions** as well.
- **Unclear presentation of methods**: There are a lot of things in the methods that are unclear; the most pressing are these:
    - What were the labels for each token? I think I understand now that the $n$th token generated by the model has the label $n$; notably, this excludes the tokens in the prompt. Maybe clarify this in the paper.
    - How were the most important individual neurons found? You say you chose them by R^2; did you train new classifiers to map from single hidden units to the length label? Or did you somehow use the previous classifiers / their R^2?
    - You say that you set the value of these neurons to value = value * scale. At which positions of the input did you do this; all of them?
- **Unclear presentation of results**: Part of the unclear presentation of the results stems from visualization choices: too many large tables are used. But there are other questions I have too:
    - In which settings does neuron scaling genuinely decrease length? From Figure 2, I gather that it works in fine-tuned settings, but only with priming in the non-fine-tuned setting. It would be helpful to plot generation length in an easier to understand way (or explain $\Delta$CR in more detail). For example, you could plot histograms / the distribution of the difference between scaled and baseline generation length, for various levels of scaling.
    - Do the human evaluations actually indicate that scaling neurons actually changes conciseness? In Table 6, conciseness of the up-scaled neuron summaries is 0.00-0.01 higher than that of the baseline, and 0.01-0.02 better than the gold standard. Is this difference in conciseness really (statistically) significant / if so, does an effect of this size really matter? And why is the gold conciseness also no better than the baseline?
- **Low engagement with prior work**: There's much to be said about probing and why it is / isn't a good idea, in particular due to its ability to decode functionally irrelevant information from model representations; this is worsened when you use MLPs rather than linear probes. Your causal interventions (maybe) validate your use of probes, but it would still be good to engage more with the probing literature; [here](https://direct.mit.edu/coli/article/48/1/207/107571/Probing-Classifiers-Promises-Shortcomings-and) is a good review paper. Similarly, [here's a review](https://direct.mit.edu/tacl/article/doi/10.1162/tacl_a_00519/113852/Neuron-level-Interpretation-of-Deep-NLP-Models-A) on neuron-level model interpretation.

**Questions:**

### Questions/Comments
- I'm a bit confused about the formulation given in (1)-(4). Most autoregressive language models in use today (including Llama-2/3 and Phi-3) use pre-LayerNorm; i.e., LayerNorm is applied on the inputs (not outputs) of each attention and MLP layer. The formulation given in (1)-(4) uses post-LayerNorm, where LayerNorm is applied to the output of each module + the residual stream. Are equations (1)-(4) wrong? Or did you modify the architectures of the pre-trained models in some way?
- Again about (1)-(4): You use $S_{emb}$ as the input to the attention module's Q/K/V vector calculations, but this should really be $x$, the current residual stream, right? Since you refer to equations (1)-(4) at layers > 0.
- Figure 1: Which model is this?
- (240-245): Please rewrite this so that it's not a string of so many questions in one block.
- (256-258): You say "This result indicates that the LLM captures length representations in the early stages, similar to how they capture semantic representations (Tenney et al., 2019; Niu et al., 2022)." but Tenney et al. stress that syntax appears in the earlier stages, while semantics appear in the latter.
- (258-259): "As such, the increase in length representations in the final layer indicates that the model revisits this information to reinforce positional context." This seems pretty speculative.
- Table 3: This is too hard to parse. Please simplify the table, by removing models / converting it into a figure that is more legible; currently  there are too many numbers/cells to read easily.
- Why do you think your method works? Though this is an interpretability paper, I'm not sure I understand why neurons that encode the current generation length should push the model to yield shorter outputs when scaled upwards. Is it because the model is aiming for a certain length, and when you scale the neurons that record that length, it acts as though the generation is already long, and cuts it short?

### Typos
- (55): coherence and informativeness for texts -> coherence and informativeness **of** texts
- (57-58, and elsewhere): multiplying positive numbers -> multiplying **them by** positive numbers

---

> ### Author Response · Authors · 2024-11-16
> **Response to questions and weakness.**
>
> >What were the labels for each token? I think I understand now that the
> th token generated by the model has the label
> ; notably, this excludes the tokens in the prompt. Maybe clarify this in the paper.
>
> Your understanding is correct. For each token generated by the model during the summary generation process, we assigned a label corresponding to its generation timestep t. We exclude the tokens in the prompt from this labeling.
>
> >How were the most important individual neurons found? You say you chose them by R^2; did you train new classifiers to map from single hidden units to the length label? Or did you somehow use the previous classifiers / their R^2?
>
> Your understanding is correct.
> To figure out the most important individual neurons, we train new classifiers to map from single hidden units to the length label.
>
> >You say that you set the value of these neurons to value = value * scale. At which positions of the input did you do this; all of them?
>
> In our experiments, we applied the scaling to the top-k individual neurons in the outputs of the second layer's attention mechanism.
>
> >In which settings does neuron scaling genuinely decrease length? From Figure 2, I gather that it works in fine-tuned settings, but only with priming in the non-fine-tuned setting. It would be helpful to plot generation length in an easier to understand way (or explain
> CR in more detail). For example, you could plot histograms / the distribution of the difference between scaled and baseline generation length, for various levels of scaling.
> The compression ratio is calculated as the ratio of the length of the generated summary to the length of the source text. The length is counted based on the number of words.
>
> Delta CR is an arithmetical difference between ratios. A delta CR value close to zero indicates that the model-generated summary has a compression ratio similar to the gold summary. The more positive delta CR means the generated summary is longer than the gold summary, while a negative delta CR means it is shorter than the gold summary.
>
> >Do the human evaluations actually indicate that scaling neurons actually changes conciseness? In Table 6, conciseness of the up-scaled neuron summaries is 0.00-0.01 higher than that of the baseline, and 0.01-0.02 better than the gold standard. Is this difference in conciseness really (statistically) significant / if so, does an effect of this size really matter? And why is the gold conciseness also no better than the baseline?
>
> We have conducted significant tests to determine whether the differences in conciseness and informativeness scores are statistically significant. We used paired-boot strap resampling with 10,000 samples [1]. We obtained the differences in conciseness and informativeness scores between scaling factors of -10 and +10 are statistically significant (p <0.05). We think this demonstrates that length-related units we found can directly control output sequence length to produce longer summaries with negative scalings or shorter summaries with positive scalings without external prompt changes. Thus, we think we can draw general conclusions that multiplying negative factors to length-related units generate longer summaries while multiplying positive factors generate shorter summaries.
>
> [1] Philipp Koehn. 2004. Statistical significance tests for machine translation evaluation. In Proceedings of the 2004 Conference on Empirical Methods in Natural Language Processing. Association for Computational Linguistics.
>
> The following is Table 6.
> Table 6
> | **Scale** |      |  **Zero-shot**       |       |    |   **Fine-tuning**     |       |
> |-----------|----------------------|-------|-------|--------------------|-------|-------|
> |           | **Coh.**            | **Conc.** | **Infor.** | **Coh.**         | **Conc.** | **Infor.** |
> | -10       | 3.73 (0.25)         | 3.56 (0.25) | **3.71**$^\dagger$ (0.27) | 3.43 (0.25) | _3.33_ (0.27) | **3.34**$^\dagger$ (0.21) |
> | 1         | 3.72 (0.29)         | 3.59 (0.23) | 3.70 (0.29) | 3.48 (0.23) | 3.46 (0.20) | 3.31 (0.27) |
> | Gold      | 3.67 (0.30)         | 3.58 (0.23) | 3.68 (0.27) | 3.42 (0.26) | 3.45 (0.22) | 3.28 (0.22) |
> | 10        | 3.69 (0.24)         | 3.59 (0.29) | _3.63_ (0.29) | 3.41 (0.22) | **3.47**$^\dagger$ (0.22) | _3.19_ (0.23) |

---

> > ### Author Response · Authors · 2024-11-16
> > **Response to questions and weakness.**
> >
> > >Low engagement with prior work: There's much to be said about probing and why it is / isn't a good idea, in particular due to its ability to decode functionally irrelevant information from model representations; this is worsened when you use MLPs rather than linear probes. Your causal interventions (maybe) validate your use of probes, but it would still be good to engage more with the probing literature; here is a good review paper. Similarly, here's a review on neuron-level model interpretation.
> >
> > We appreciate introducing related papers. However, we believe that our experimental results demonstrate that casual interventions successfully generate shorter or longer summaries based on the length-related units we identified. We have carefully read the papers and included such probing papers to improve our draft.
> >
> > >I'm a bit confused about the formulation given in (1)-(4). Most autoregressive language models in use today (including Llama-2/3 and Phi-3) use pre-LayerNorm; i.e., LayerNorm is applied on the inputs (not outputs) of each attention and MLP layer. The formulation given in (1)-(4) uses post-LayerNorm, where LayerNorm is applied to the output of each module + the residual stream. Are equations (1)-(4) wrong? Or did you modify the architectures of the pre-trained models in some way?
> >
> > We appreciate your careful reading to improve our drafts.
> >
> > > Figure 1: Which model is this?
> >
> > We used the Llama-2-7B-Chat model in line 250
> >
> > >(240-245): Please rewrite this so that it's not a string of so many questions in one block.
> >
> > We appreciate your careful reading. We will revise our draft based on your suggestion.
> >
> > >(256-258): You say "This result indicates that the LLM captures length representations in the early stages, similar to how they capture semantic representations (Tenney et al., 2019; Niu et al., 2022)." but Tenney et al. stress that syntax appears in the earlier stages, while semantics appear in the latter.
> >
> > We appreciate your careful reading. As Tenney stresses that semantics appear in the latter, the recent work of Niu shows that both syntactic and semantic representations are captured in the bottom. We apologize for misleading the citations and will describe details in the draft.
> >
> > >(258-259): "As such, the increase in length representations in the final layer indicates that the model revisits this information to reinforce positional context." This seems pretty speculative.
> >
> > We appreciate your suggestion. We will tone down the sentence.
> >
> > >Why do you think your method works? Though this is an interpretability paper, I'm not sure I understand why neurons that encode the current generation length should push the model to yield shorter outputs when scaled upwards. Is it because the model is aiming for a certain length, and when you scale the neurons that record that length, it acts as though the generation is already long, and cuts it short?
> >
> > Thank you for your insightful question regarding the rationale behind our method.
> > The model attempts to summarize input to a desired length based on length-constraint prompts. We believe when we scale certain length-related units that record the current generation length, it effectively signals to the model that the generated summary is already longer than desired.

---

> ### Comment · Reviewer_KeXg · 2024-11-17
>
> Hi! Thanks for your responses. Some of them answered my questions, and some did not, so I'll repeat / follow-up on those here:
>
> > You say that you set the value of these neurons to value = value * scale. At which positions of the input did you do this; all of them?
>
> > In our experiments, we applied the scaling to the top-k individual neurons in the outputs of the second layer's attention mechanism.
>
> Here, I was referring to position in the sentence, not within the transformer. Is the scaling applied at all token positions?
>
> > In which settings does neuron scaling genuinely decrease length? ...
>
> > Delta CR is an arithmetical difference between ratios. A delta CR value close to zero indicates that the model-generated summary has a compression ratio similar to the gold summary. The more positive delta CR means the generated summary is longer than the gold summary, while a negative delta CR means it is shorter than the gold summary.
>
> Thanks for explaining Delta CR, but I asked about the settings in which neuron scaling genuinely decreases length. I asked this because, in Figure 2, it's not clear that it does work in all scenarios. It seems that either fine-tuning or prompting length-specific prompting is necessary to make it work, which has implications for this method's utility, and the conclusions we should draw about model mechanisms (e.g. do they use different mechanisms when prompted or fine-tuned, yielding causal interventions to only be effective in some cases?).
>
> > Do the human evaluations actually indicate that scaling neurons actually changes conciseness?... Is this difference in conciseness really (statistically) significant / if so, does an effect of this size really matter?...
>
> > We have conducted significant tests to determine whether the differences in conciseness and informativeness scores are statistically significant. We used paired-boot strap resampling with 10,000 samples [1]. We obtained the differences in conciseness and informativeness scores between scaling factors of -10 and +10 are statistically significant (p <0.05). We think this demonstrates that length-related units we found can directly control output sequence length to produce longer summaries with negative scalings or shorter summaries with positive scalings without external prompt changes. Thus, we think we can draw general conclusions that multiplying negative factors to length-related units generate longer summaries while multiplying positive factors generate shorter summaries.
>
> Thanks for this response! Two things remain slightly unclear to me. First, is it the case that only the difference in conciseness between -10 and +10 scaling is statistically significant (and other pairs, e.g. +0 vs. +10 are not significant)? Second, as I asked originally, and irrespective of statistical significance, what do you think of the effect size? It seems quite small to me, so is this really an impactful intervention?
>
> > I'm a bit confused about the formulation given in (1)-(4)...Are equations (1)-(4) wrong? Or did you modify the architectures of the pre-trained models in some way?
>
> > We appreciate your careful reading to improve our drafts.
>
> To be clear, equations (1)-(4) are wrong then, and you use a standard pre-LayerNorm architecture?

---

> > ### Author Response · Authors · 2024-11-24
> >
> > Thank you very much for your feedback to improve our draft!
> >
> > >Here, I was referring to position in the sentence, not within the transformer. Is the scaling applied at all token positions?
> >
> > We apologize for the confusion. We scaled the hidden units by multiplying them with positive or negative numeric values to adjust the output sequence length. This scaling was applied to all output token positions during the generation process, excluding the input token prompts.
> >
> > >Thanks for explaining Delta CR, but I asked about the settings in which neuron scaling genuinely decreases length. I asked this because, in Figure 2, it's not clear that it does work in all scenarios. It seems that either fine-tuning or prompting length-specific prompting is necessary to make it work, which has implications for this method's utility, and the conclusions we should draw about model mechanisms (e.g. do they use different mechanisms when prompted or fine-tuned, yielding causal interventions to only be effective in some cases?).
> >
> > Your understanding is correct. Our method is more effective when the input prompt is length-specific. For example, the LLM using a no-constraint prompt does not effectively control the output sequence length with our causal intervention. We believe this is because the prompt itself does not include length-specific constraints.
> >
> > However, length-specific and priming prompts in both zero-shot and fine-tuning settings demonstrate better results. When fine-tuned with length-specific prompts, these length-related units become more active. Therefore, we demonstrate that our method is effective when length-specific prompts are provided in zero-shot settings or when the model is fine-tuned.
> >
> > >Thanks for this response! Two things remain slightly unclear to me. First, is it the case that only the difference in conciseness between -10 and +10 scaling is statistically significant (and other pairs, e.g. +0 vs. +10 are not significant)? Second, as I asked originally, and irrespective of statistical significance, what do you think of the effect size? It seems quite small to me, so is this really an impactful intervention?
> >
> > We appreciate your thoughtful questions! Let me clarify these points.
> > Regarding statistical significance, it is indeed the case that the difference in conciseness between -10 and +10 scaling is statistically significant, whereas the pair between +0  and +10 does not show significance. However, we think this highlights the critical role of extreme scaling values in effectively controlling output length.
> >
> > Regarding the effect size, while it may appear small in absolute terms, we believe it is still impactful. Previous works demonstrated that ensuring correct output length often required considerable effort and significant model modifications in statistical machine translation systems [1]. Similarly, our findings show that length control remains challenging, even with state-of-the-art systems like InstructCMP, which struggle to achieve precise length control without fine-tuning or specialized prompts [2]. We believe our work sheds light on how causal interventions, while not universally effective, can contribute to improving control under specific conditions and to understanding the inner mechanisms of LLMs.
> >
> > [1] Xing shi, Why Neural Translations are the Right Length. Proceedings of the 2016 Conference on Empirical Methods in Natural Language Processing.
> >
> > [2] Do, 2024. InstructCMP: Length Control in Sentence Compression through Instruction-based Large Language Models, Findings of the Association for Computational Linguistics: ACL 2024.
> >
> > > To be clear, equations (1)-(4) are wrong then, and you use a standard pre-LayerNorm architecture?
> >
> > We are sorry for confusion. You are correct. We used a standard pre-LayerNorm architecture. We uploaded modified draft.

---

> > > ### Comment · Reviewer_KeXg · 2024-11-24
> > >
> > > Thanks for this response—it clears things up for me a bit. Ultimately, large error bars in Figures 2-3, and the small (though sometimes significant) effect size in the human experiments leave me doubting the real efficacy of this technique. Moreover, there's just not quite enough interpretability work done in this paper to actually explain why this technique might work; this could have justified the paper's acceptance despite the aforementioned issues. I will keep my score the same, but thanks for your engagement throughout this discussion period.

---

### Official Review · Reviewer_fPnc · 2024-11-01

**Soundness:** 1
**Presentation:** 2
**Contribution:** 2
**Rating:** 3
**Confidence:** 4

**Summary:**

This paper studies the internal mechanisms of how LLMs control the length of the generated output sequences. It presents experiments on a summarization task where information on sequence length is probed across several components of the transformer architecture, with findings indicating that the attention heads from lower layers hold such information. The paper also suggests that such information is stored in a “disentangled” manner, so that the length can be tuned without sacrificing the informativeness of the generated text. Finally, the paper presents a human evaluation to qualitatively confirm its findings.

**Strengths:**

Overall I think studying how length can be controlled through internal representations is an interesting problem with potentially useful applications, particularly if it can lead to insights on how prompt instructions relate to internal representations.

I also appreciate that this study uses several different LLMs, ensuring that the findings would generalize across models.

**Weaknesses:**

As far as I can tell, several of the claims made in the paper are not backed up by the results. Furthermore, the paper lacks statistical reporting so it is hard to assess whether the findings are significant. I am also not convinced that the experimental methodology is sound; however, that could potentially be due to misunderstandings on my part that can be addressed by improved clarity in a future draft. See the bullet points below for detailed feedback.

1. 3.1: (a) you claim that the “length” prompt instructs the model to provide a summary with a specific desired length. However, that is not what the prompt in Table 1 specifies, it specifies a number of words that should be *deleted*. Even for that, I am not sure it shaves off the *least* important words, which I inferred was the purpose. (b) For the “priming” prompt, wouldn’t the two user-provided numbers work against each other? I could specify that I want a summary of, say, 10 words and also specify that I want to delete some number of words; but deleting that many may not yield a sentence with exactly 10 words. (c) In light of these inconsistencies, did you look at whether the model outputs actually adhered to the objectives you intended? (d) Overall, I am not convinced that these particular prompts are sufficient to draw general conclusions about how the instructions given in the prompt interact with the parts of the network that are activated (as is done, e.g., in 5.1).
2. I have some concerns about your method to find evidence of length representations (end of 3.1). (a) You use a neural network, so it is misleading to call it linear regression. The text also lacks details on how you fit the parameters of this model. (b) It is not clear to me how the response variable “generation time step” is defined. Does it include the prompt? The variable n is also not very well specified. (c) It looks like you’re predicting the “generation time step” from a matrix of hidden states. However, couldn’t that just be inferred by the attention masking or the positional embeddings?
3. Figure 2: (a) I think some of your claims in 5.2 are not supported by the data. Looking at the rogue scores in Figure 2, there is very little positive difference. The plot is also missing confidence intervals, so I cannot assess whether the differences are within the error margin. This is a more general issue—none of the results in the paper are presented with error bars. (b) I also think the term “disentangled” is misleading here: Looking at the plot, there does seem to be an interaction effect between output length (delta CR) and quality (Rogue-L).
4. Human evaluations: There are no error margins reported here either. Since the scores presented in Table 6 are very close to each other, I am not convinced that you can draw any conclusions based on those results.

**Questions:**

1. What is the rationale for studying fine-tuned models, particularly using QLoRA? Why are you interested in 4 vs. 8-bit quantization?

2. 3.2: This looks like probing—there is much literature on that. How does your method relate to probing methods?

---

> ### Author Response · Authors · 2024-11-16
> **Response to questions and weakness.**
>
> > 3.1: (a) you claim that the “length” prompt instructs the model to provide a summary with a specific desired length. However, that is not what the prompt in Table 1 specifies, it specifies a number of words that should be deleted. Even for that, I am not sure it shaves off the least important words, which I inferred was the purpose. (b) For the “priming” prompt, wouldn’t the two user-provided numbers work against each other? I could specify that I want a summary of, say, 10 words and also specify that I want to delete some number of words; but deleting that many may not yield a sentence with exactly 10 words. (c) In light of these inconsistencies, did you look at whether the model outputs actually adhered to the objectives you intended? (d) Overall, I am not convinced that these particular prompts are sufficient to draw general conclusions about how the instructions given in the prompt interact with the parts of the network that are activated (as is done, e.g., in 5.1).
>
> For the "Length" prompt, we believe it effectively instructs and guides LLMs to control output sequence length to meet desired lengths in that the prompt specifies the number of words to be deleted. LLMs can infer the intended length information through in-context learning. Additionally, previous studies have confirmed that significant improvements in ROUGE scores when using the "Length" prompt compared to "No-constraint" prompts [1]. It demonstrates that the "Length" prompt helps the model to preserve important words while controlling the output length.
>
> We believe that the "priming" prompt is suitable and adheres to the objectives [1]. Specifically, as shown in Table  5, the Base method using the "Priming" prompt with fine-tuning setting consistently generates summaries closer to gold summary lengths in bin-wise analysis. Based on our results, we are confident that used prompts are sufficient to draw general conclusions about how instructions given in the prompt interact with the parts of the networks that are activated. The identified length-related hidden units further support our findings in that multiplying positive factors generates shorter summaries while multiplying negative factors generates longer summaries.
>
> [1] Do, 2024. InstructCMP: Length Control in Sentence Compression through Instruction-based Large Language Models, Findings of the Association for Computational Linguistics: ACL 2024.
>
> > I have some concerns about your method to find evidence of length representations (end of 3.1).
> (a) You use a neural network, so it is misleading to call it linear regression. The text also lacks details on how you fit the parameters of this model.
>
> Your understanding is correct. However, we think the text itself is detailed for explaining how we fit the parameters of this model as we have described the dimensions with formulas.
>
> > (b) It is not clear to me how the response variable “generation time step” is defined. Does it include the prompt? The variable n is also not very well specified.
>
> For each token generated by the model during the summary generation process, we assigned a label corresponding to its generation timestep. We exclude the tokens in the prompt from this labeling.
>
> > (c) It looks like you’re predicting the “generation time step” from a matrix of hidden states. However, couldn’t that just be inferred by the attention masking or the positional embeddings?
>
> We appreciate your insightful suggestion. While investigating attention masking or positional embeddings would provide valuable insights, these approaches are beyond the scope of our current study. Moreover, as we conducted experiments using all hidden units in Figure 1, the first layer with attention-residual outputs does not show a higher R² compared to other layers. Thus, simply investigating positional embeddings might not be as suitable as analyzing entire layers in LLMs. In addition, we believe that investigating word embeddings themselves is more appropriate since certain words, such as ``I’’ or periods, are frequently located at the beginning or end of sentences. Thus, they may inherently contain relevant information. In the future, we will explore these aspects including your suggestions to further understanding of LLMs internal mechanisms.
>
> > Figure 2: (a) I think some of your claims in 5.2 are not supported by the data. ... differences are within the error margin. This is a more general issue—none of the results in the paper are presented with error bars.
>
> Our goal is to identify the length-related hidden units in LLMs and control output sequence length by manipulating those units. As you commented the Rouge-L has very little positive difference meaning that multiplying length-related hidden units can generate shorter or longer summaries without losing informativeness. We will add the error margin as soon as possible. Thank you very much.

---

> ### Author Response · Authors · 2024-11-16
> **Response to questions and weakness.**
>
> For all Tables and Figures, we will add error bars with standard deviations.
> The following is Table 2 including standard deviation. Please note that standard deviations for the numbers in Table 3 are also nearly zero.
>
>  **Model**                  | **Constraint**   | **Attn Out (F)** | **Attn Out (S)** | **Attn Out (L)** | **Attn Residual (F)** | **Attn Residual (S)** | **Attn Residual (L)** | **MLP Out (F)** | **MLP Out (S)** | **MLP Out (L)** | **MLP Residual (F)** | **MLP Residual (S)** | **MLP Residual (L)** |
> |----------------------------|------------------|------------------|------------------|------------------|------------------------|-----------------------|-----------------------|-----------------|-----------------|-----------------|-----------------------|-----------------------|-----------------------|
> | **Llama-2-7B**            | **No-constraint** | 0.94 (0.00)     | **0.95 (0.00)**  | 0.88 (0.00)      | 0.00 (0.00)           | 0.90 (0.00)          | 0.89 (0.00)          | 0.84 (0.00)    | 0.70 (0.00)    | 0.67 (0.01)    | 0.90 (0.00)           | 0.94 (0.00)           | 0.85 (0.00)          |
> |                            | **Length**       | 0.98 (0.00)     | **0.99 (0.00)**  | 0.93 (0.00)      | 0.11 (0.00)           | 0.94 (0.00)          | 0.93 (0.00)          | 0.89 (0.00)    | 0.77 (0.00)    | 0.70 (0.01)    | 0.95 (0.00)           | 0.97 (0.00)           | 0.89 (0.00)          |
> |                            | **Priming**      | 0.98 (0.00)     | **0.99 (0.00)**  | 0.95 (0.00)      | 0.11 (0.00)           | 0.94 (0.00)          | 0.94 (0.00)          | 0.89 (0.00)    | 0.77 (0.00)    | 0.78 (0.00)    | 0.95 (0.00)           | 0.98 (0.00)           | 0.92 (0.00)          |
> | **Llama-2-13B**           | **No-constraint** | 0.95 (0.00)     | **0.96 (0.00)**  | 0.93 (0.00)      | 0.08 (0.01)           | 0.93 (0.00)          | 0.92 (0.00)          | 0.90 (0.00)    | 0.83 (0.00)    | 0.74 (0.01)    | 0.93 (0.00)           | 0.95 (0.00)           | 0.89 (0.00)          |
> |                            | **Length**       | **0.94 (0.00)** | **0.94 (0.00)**  | 0.92 (0.00)      | 0.10 (0.00)           | 0.92 (0.00)          | 0.92 (0.00)          | 0.89 (0.00)    | 0.81 (0.00)    | 0.75 (0.00)    | 0.91 (0.00)           | **0.94 (0.00)**       | 0.91 (0.00)          |
> |                            | **Priming**      | **0.99 (0.00)** | **0.99 (0.00)**  | 0.91 (0.00)      | 0.17 (0.00)           | 0.96 (0.00)          | 0.92 (0.00)          | 0.92 (0.00)    | 0.81 (0.00)    | 0.72 (0.00)    | 0.97 (0.00)           | 0.98 (0.00)           | 0.89 (0.00)          |
> | **Llama-2-70B**           | **No-constraint** | 0.97 (0.00)     | **0.99 (0.00)**  | 0.95 (0.00)      | 0.16 (0.00)           | 0.93 (0.00)          | 0.92 (0.00)          | 0.83 (0.01)    | 0.81 (0.01)    | 0.82 (0.00)    | 0.95 (0.00)           | 0.98 (0.00)           | 0.92 (0.00)          |
> |                            | **Length**       | 0.97 (0.00)     | **0.99 (0.00)**  | 0.94 (0.00)      | 0.17 (0.00)           | 0.92 (0.00)          | 0.93 (0.00)          | 0.87 (0.00)    | 0.84 (0.00)    | 0.80 (0.01)    | 0.95 (0.00)           | 0.98 (0.00)           | 0.92 (0.00)          |
> |                            | **Priming**      | **0.98 (0.00)** | 0.97 (0.00)      | 0.91 (0.00)      | 0.18 (0.00)           | 0.91 (0.00)          | 0.89 (0.00)          | 0.82 (0.00)    | 0.76 (0.01)    | 0.78 (0.01)    | 0.94 (0.00)           | 0.95 (0.00)           | 0.88 (0.00)          |
> | **Llama-3-8B**            | **No-constraint** | 0.96 (0.00)     | **0.98 (0.00)**  | 0.91 (0.00)      | 0.20 (0.00)           | 0.86 (0.00)          | 0.91 (0.00)          | 0.70 (0.00)    | 0.74 (0.00)    | 0.78 (0.01)    | 0.88 (0.00)           | 0.95 (0.00)           | 0.88 (0.00)          |
> |                            | **Length**       | 0.96 (0.00)     | **0.97 (0.00)**  | 0.93 (0.00)      | 0.16 (0.00)           | 0.88 (0.00)          | 0.93 (0.00)          | 0.72 (0.00)    | 0.75 (0.00)    | 0.79 (0.01)    | 0.90 (0.00)           | 0.96 (0.00)           | 0.89 (0.00)          |
> |                            | **Priming**      | 0.97 (0.00)     | **0.98 (0.00)**  | 0.94 (0.00)      | 0.24 (0.00)           | 0.87 (0.00)          | 0.94 (0.00)          | 0.73 (0.00)    | 0.76 (0.00)    | 0.87 (0.00)    | 0.89 (0.00)           | 0.95 (0.00)           | 0.92 (0.00)          |

---

> > ### Author Response · Authors · 2024-11-16
> > **Response to questions and weakness.**
> >
> > The following is Table 2 with errors.
> >
> >
> >
> >  **Model**                  | **Constraint**   | **Attn Out (F)** | **Attn Out (S)** | **Attn Out (L)** | **Attn Residual (F)** | **Attn Residual (S)** | **Attn Residual (L)** | **MLP Out (F)** | **MLP Out (S)** | **MLP Out (L)** | **MLP Residual (F)** | **MLP Residual (S)** | **MLP Residual (L)** |
> > |----------------------------|------------------|------------------|------------------|------------------|------------------------|-----------------------|-----------------------|-----------------|-----------------|-----------------|-----------------------|-----------------------|-----------------------|
> > | **Phi3-mini-4k**          | **No-constraint** | 0.93 (0.00)     | **0.97 (0.00)**  | 0.91 (0.00)      | 0.07 (0.01)           | 0.80 (0.01)          | 0.91 (0.00)          | 0.61 (0.01)    | 0.66 (0.01)    | 0.55 (0.01)    | 0.84 (0.00)           | 0.95 (0.00)           | 0.86 (0.00)          |
> > |                            | **Length**       | 0.94 (0.00)     | **0.97 (0.00)**  | 0.92 (0.00)      | 0.04 (0.00)           | 0.80 (0.00)          | 0.92 (0.00)          | 0.65 (0.01)    | 0.67 (0.00)    | 0.56 (0.01)    | 0.82 (0.00)           | 0.94 (0.00)           | 0.86 (0.00)          |
> > |                            | **Priming**      | 0.93 (0.00)     | **0.97 (0.00)**  | 0.89 (0.00)      | 0.07 (0.01)           | 0.77 (0.00)          | 0.90 (0.00)          | 0.48 (0.01)    | 0.63 (0.01)    | 0.58 (0.01)    | 0.80 (0.00)           | 0.95 (0.00)           | 0.84 (0.00)          |
> > | **Phi3-small-8k**         | **No-constraint** | 0.92 (0.01)     | **0.95 (0.00)**  | 0.71 (0.01)      | 0.01 (0.01)           | 0.83 (0.00)          | 0.83 (0.00)          | 0.76 (0.01)    | 0.72 (0.01)    | 0.48 (0.00)    | 0.87 (0.00)           | 0.90 (0.00)           | 0.81 (0.01)          |
> > |                            | **Length**       | 0.94 (0.00)     | **0.97 (0.00)**  | 0.85 (0.00)      | 0.11 (0.01)           | 0.86 (0.00)          | 0.87 (0.00)          | 0.80 (0.00)    | 0.79 (0.01)    | 0.54 (0.01)    | 0.89 (0.00)           | 0.94 (0.00)           | 0.87 (0.00)          |
> > |                            | **Priming**      | 0.97 (0.00)     | **0.98 (0.00)**  | 0.81 (0.01)      | 0.27 (0.01)           | 0.92 (0.00)          | 0.87 (0.01)          | 0.86 (0.01)    | 0.83 (0.00)    | 0.58 (0.00)    | 0.93 (0.00)           | 0.97 (0.00)           | 0.86 (0.00)          |

---

> > > ### Author Response · Authors · 2024-11-16
> > > **Response to questions and weakness.**
> > >
> > > The following is Table 5 with the standard deviation. Compared to Base which does not multiply factors, scaling +10 or -10 successfully generates shorter or longer summaries. Dagger indicates the statistical significance (p<0.05) using paired boot-strapping methods with 10,000 samples [1].
> > >
> > > [1] Philipp Koehn. 2004. Statistical significance tests for machine translation evaluation. In Proceedings of the 2004 Conference on Empirical Methods in Natural Language Processing. Association for Computational Linguistics.
> > >
> > > | **Setting**    | **Prompting**   | **Scale** | **1-5**      | **6-10**     | **11-15**     | **16-20**     | **21-25**     | **25-30**     | **30-**      |
> > > |----------------|-----------------|-----------|--------------|--------------|---------------|---------------|---------------|---------------|--------------|
> > > | **Zero-shot**  | **Length**      | **#Data** | 2            | 91           | 275           | 282           | 177           | 102           | 71           |
> > > |                |            | **Base**  | -15.31 (15.3) / 50.00 (50.0) | 4.65 (15.8) / 78.30 (22.1) | 19.28 (17.06) / 73.64 (17.1) | 33.53 (15.7) / 65.53 (14.8) | 45.37 (15.1) / 56.29 (13.8) | 53.31 (12.2) / 49.39 (13.2) | 59.19 (11.5) / 43.93 (13.4) |
> > > |                |                 | **10**    | -15.31 (15.3) / 50.00 (50.0) | 10.38 (20.1) / 76.54 (21.7) | 22.44 (18.0) / 73.25 (16.7) | 34.53 (18.9) / 65.01 (16.0) | 46.81 (16.9) / 55.70 (14.4) | 52.71 (15.1) / 50.21 (14.6) | 60.26 (11.0) / 44.04 (13.5) |
> > > |                |                 | **-10**   | -15.31 (15.3) / 50.00 (50.0) | 20.62 (23.7)$^\dagger$ / 73.45 (20.7) | 33.58 (19.9)$^\dagger$ / 67.68 (16.5) | 43.09 (20.6)$^\dagger$ / 62.33 (16.6) | 55.65 (15.4)$^\dagger$ / 53.23 (14.1) | 55.93 (17.2)$^\dagger$ / 48.78 (14.5) | 63.98 (13.9)$^\dagger$ / 43.30 (13.3) |
> > > |                | **Priming**     | **#Data** | 33           | 324          | 382           | 176           | 57            | 24            | 4            |
> > > |                |                 | **Base**  | 0.71 (5.6) / 79.35 (25.3) | 3.22 (10.1) / 74.33 (24.3) | 11.82 (11.9) / 74.02 (18.6) | 22.03 (13.5) / 69.38 (17.4) | 35.66 (14.0) / 61.83 (15.0) | 39.35 (14.5) / 57.24 (13.1) | 44.90 (4.7) / 53.52 (17.8) |
> > > |                |                 | **10**    | 0.81 (5.4) / 80.65 (24.6) | 1.58 (10.6)$^\dagger$ / 74.79 (23.7) | 9.89 (15.1)$^\dagger$ / 73.92 (19.6) | 17.30 (18.9)$^\dagger$ / 71.49 (18.6)$^\dagger$ | 24.60 (21.1)$^\dagger$ / 64.60 (21.0) | 36.13 (19.0) / 60.77 (14.8) | 36.69 (20.4) / 63.91 (23.5) |
> > > |                |                 | **-10**   | 1.36 (13.0) / 61.89 (37.1) | 13.88 (22.7)$^\dagger$ / 66.25 (26.8) | 24.06 (24.4)$^\dagger$ / 64.38 (23.0) | 30.78 (23.2)$^\dagger$ / 63.02 (20.9) | 42.78 (20.5)$^\dagger$ / 58.57 (16.6) | 39.48 (22.1) / 57.42 (17.5) | 50.29 (8.3) / 56.60 (9.5) |
> > > | **Fine-tuning**| **Length**      | **#Data** | 28           | 464          | 417           | 82            | 7             | 2             | 0            |
> > > |                |                 | **Base**  | -2.12 (4.9) / 79.17 (30.1) | 1.34 (9.5) / 85.07 (18.1) | 6.97 (10.9) / 82.99 (17.1) | 13.01 (13.3) / 83.63 (12.5) | 19.85 (8.2) / 84.08 (7.7) | 38.65 (10.4) / 57.05 (26.3) | --           |
> > > |                |                 | **10**    | -1.68 (4.9) / 80.83 (29.3) | 1.26 (9.3) / 85.10 (18.1) | 4.41 (10.6)$^\dagger$ / 83.58 (17.9) | 5.42 (14.0)$^\dagger$ / 82.01 (16.9) | 10.37 (7.3) / 82.47 (20.6) | 26.83 (1.4) / 61.67 (21.7) | --           |
> > > |                |                 | **-10**   | 0.08 (8.5)$^\dagger$ / 73.22 (30.2) | 3.69 (11.8)$^\dagger$ / 83.26 (17.9) | 9.06 (12.4)$^\dagger$ / 82.45 (16.8) | 11.51 (14.7) / 81.52 (15.0) | 18.59 (9.5) / 85.07 (8.5) | 40.47 (12.3) / 58.74 (24.6) | --           |
> > > |                | **Priming**     | **#Data** | 64           | 565          | 321           | 49            | 1             | 0             | 0            |
> > > |                |                 | **Base**  | 0.07 (2.7) / 87.98 (25.0) | -0.41 (4.2) / 86.36 (20.0) | 0.06 (5.6) / 83.86 (18.6) | 1.00 (6.0) / 83.59 (15.3) | 0.00 (0.0) / 83.72 (0.0) | --           | --           |
> > > |                |                 | **10**    | -0.65 (3.5)$^\dagger$ / 87.47 (25.8) | -3.18 (5.1)$^\dagger$ / 84.86 (19.8) | -5.29 (6.7)$^\dagger$ / 81.38 (17.9) | -7.71 (6.7)$^\dagger$ / 80.34 (15.8) | -8.70 (0.0)$^\dagger$ / 78.05 (0.0) | --           | --           |
> > > |                |                 | **-10**   | 3.21 (5.2)$^\dagger$ / 81.36 (24.7) | 9.78 (8.8)$^\dagger$ / 78.59 (18.3) | 17.29 (9.7)$^\dagger$ / 77.47 (13.4) | 16.92 (10.2)$^\dagger$ / 79.22 (12.3) | 15.22 (0.0)$^\dagger$ / 84.00 (0.0) | --           | --           |

---

> > > > ### Author Response · Authors · 2024-11-16
> > > > **Response to questions and weakness.**
> > > >
> > > > The following is Table 6 with the same notation with previously answered statistically significant test settings.
> > > > The differences between -10 and 10 are statistically significant.
> > > >
> > > > Table 6
> > > > | **Scale** |      |  **Zero-shot**       |       |    |   **Fine-tuning**     |       |
> > > > |-----------|----------------------|-------|-------|--------------------|-------|-------|
> > > > |           | **Coh.**            | **Conc.** | **Infor.** | **Coh.**         | **Conc.** | **Infor.** |
> > > > | -10       | 3.73 (0.25)         | 3.56 (0.25) | **3.71**$^\dagger$ (0.27) | 3.43 (0.25) | _3.33_ (0.27) | **3.34**$^\dagger$ (0.21) |
> > > > | 1         | 3.72 (0.29)         | 3.59 (0.23) | 3.70 (0.29) | 3.48 (0.23) | 3.46 (0.20) | 3.31 (0.27) |
> > > > | Gold      | 3.67 (0.30)         | 3.58 (0.23) | 3.68 (0.27) | 3.42 (0.26) | 3.45 (0.22) | 3.28 (0.22) |
> > > > | 10        | 3.69 (0.24)         | 3.59 (0.29) | _3.63_ (0.29) | 3.41 (0.22) | **3.47**$^\dagger$ (0.22) | _3.19_ (0.23) |

---

> > > > > ### Author Response · Authors · 2024-11-16
> > > > > **Response to questions and weakness.**
> > > > >
> > > > > >What is the rationale for studying fine-tuned models, particularly using QLoRA? Why are you interested in 4 vs. 8-bit quantization?
> > > > >
> > > > > Because our goal is to investigate whether the length-related layers remain consistent regardless of QLoRA fine-tuning settings and quantization configurations, our experimental results support that they are consistent within the same layer, specifically in the second layer’s attention outputs.
> > > > >
> > > > > >3.2: This looks like probing—there is much literature on that. How does your method relate to probing methods?
> > > > >
> > > > > Our method shares the goal of understanding model representations with probing methods. However, the representations identified by probing methods do not guarantee LLMs actually use these representations for length control [2], we further investigated this by multiplying factors.
> > > > >
> > > > > [2] Ravichander. 2021. Probing the Probing Paradigm: Does Probing Accuracy Entail Task Relevance?. In Proceedings of the 16th Conference of the European Chapter of the Association for Computational Linguistics: Main Volume.

---

> > > > > > ### Comment · Reviewer_fPnc · 2024-11-24
> > > > > >
> > > > > > Thank you for engaging in discussion. However, most of my concerns have not been addressed.
> > > > > >
> > > > > > I remain unconvinced about the prompt instructions, they seem to be inconsistent with what is claimed about them in the text. This is independent of empirical results pointing in the direction that these particular instructions yield summaries with good ROGUE scores, or previous papers that may have used these prompts.
> > > > > >
> > > > > > > In the future, we will explore these aspects including your suggestions to further understanding of LLMs internal mechanisms.
> > > > > >
> > > > > > Thank you for engaging with this point, but it was not meant as a suggestion for future work. It’s a question to assess the soundness of the approach. My concern is whether it is even a well-posed task to predict the generation time step from the attention outputs. My suspicion is that I can apply a very simple prediction rule that takes the number of unmasked representations and subtracts the number of tokens in the prompt to get the generation time step. And indeed, the R^2 values reported in Table 2 are very high.
> > > > > >
> > > > > > > We will add the error margin as soon as possible.
> > > > > >
> > > > > > It’s great that you added error margins. We now see that nearly all results are within the error margins, which means that we cannot draw any conclusions based on these plots. However, you seem to not have revised your claims in this section. (Moreover, it seems like you are reporting standard deviations rather than standard errors, which would give very conservative error bars. Consider making a Gaussian assumption and compute CIs accordingly. Or alternatively, consider using some nonparametric method like bootstrap sampling.)
> > > > > >
> > > > > > Moreover, in other places like section 5.4, the results are discussed in the same manner as before, despite the fact that most of the differences appear to be statistically non-significant. In addition, I do not see any explanation on how these hypothesis tests were carried out.
> > > > > >
> > > > > > Many of the minor points I made remain unaddressed in the draft as well, like specifiying how the regressor network was trained and discussing the approach in relation to probing methods.
> > > > > >
> > > > > > I maintain my recommendation to reject this paper.

---

### Official Review · Reviewer_Khqj · 2024-11-04

**Soundness:** 2
**Presentation:** 2
**Contribution:** 3
**Rating:** 3
**Confidence:** 4

**Summary:**

This work investigates where output length is linearly encoded in an LM's hidden activations. In learning linear regressions from hidden units to the output length, it is found that length is encoded in early layers of the model. Finally, when the specific hidden units encoding length are scaled, the model output changes accordingly without sacrificing grammaticality.

**Strengths:**

Overall, this paper addresses an important problem with thorough experiments. However, as it stands, there were a number of issues with the presentation that decrease its correctness and potential impact (see weaknesses).

1. __Originality__ This work constitutes the first thorough experimental work on length representations in LLMs, as far as I'm aware. Although the experimental methods are not novel, their application to output length representation is.
2. __Quality__ The experimental work is broad (but only partially reported, see weaknesses).
3. __Clarity__ The paper is overall understandable and engaging to read. There are several aspects of the writing and methodology, especially regarding reproducibility, that can be improved, see weaknesses.
4. __Significance__ The problem of length representations in LMs is a useful one. The paper shows that it is, moreover, possible to tune the length in a fine-grained way without sacrificing the semantics of the output.

**Weaknesses:**

While the goals of the paper are very promising, I found the execution and presentation to be quite unclear, which lowers the contribution's potential impact. In particular, weaknesses 1-4, and especially 3, lowered my score (weakness 5 did not impact my score).

1. __Finetuning methodology__ I had trouble understanding the motivation and the details for finetuning (after reading Appendix A).
	1. What data were the models fine-tuned on?
        2. Why do we want to finetune the models? What does this tell us in addition to the zero-shot setting?
2. __Evaluation methodology unclear__
	1. How were models sampled? E.g. greedy decoding, top-k?
	2. As LMs have variable-length inputs, which tokens' representations are being used for the linear regressions?
	3. l402: Please include a summary of how $\Delta CR$ should be interpreted; please define compression ratio in the text.
	4. Table 4: "The numbers in parentheses indicate an index of hidden units from the second layer of the attention mechanisms." I did not understand this; how many hidden units are there total? Are the reported values for a single hidden unit, or averaged over several individual regressions?
3. __Partial results reported in manuscript__ While Section 3 (Methods) states that experiments are on Llama 2, Llama 3, and Phi, only Section 4 reports results for all models. The rest of the paper only reports results on Llama 2. Please include the full set of results for Section 5.
4. __No statistical significance / random seeds reported__ The lack of significance reporting makes it very difficult to quantitatively evaluate the results. In particular:
	1. Table 6 (human evaluations). It is impossible to tell whether the differences between -10 and 10 are statistically significant without errors. Please include errors and significance values.
	2. Table 5: N is reported in the # data row, but are the values an average over N? If so, can you please report the standard deviation?
        3. Figures: please report error bars with standard deviations over random seeds.
5. Minor presentation: In Figs 1-3 please scale the y-axes to the same range.

Concretely, I would increase my rating if weaknesses 1-4 are addressed, with an emphasis on 3 and 4.

**Questions:**

See weaknesses

---

> ### Author Response · Authors · 2024-11-16
> **Response to questions and weakness.**
>
> >Finetuning methodology I had trouble understanding the motivation and the details for finetuning (after reading Appendix A).
> >What data were the models fine-tuned on?
>
> The models were fine-tuned on the Google sentence summarization dataset, which consists of instruction-based formats, No-constraint, Length, Priming in Table 1 [1]. The fine-tuning process is supervised by incorporating length-specific instructions to enhance the model’s ability to control the output sequence length.
>
> [1] Do, 2024. InstructCMP: Length Control in Sentence Compression through Instruction-based Large Language Models, Findings of the Association for Computational Linguistics: ACL 2024.
>
> >Why do we want to finetune the models? What does this tell us in addition to the zero-shot setting?
>
> By fine–tuning the models with length-specific prompts, we target to reinforce their length-control capabilities. Fine-tuning allows the models to learn length constraints more robustly compared to relying solely on zero-shot settings.
>
> In addition, we aim to investigate how explicit fine-tuning with length constraint prompts affect length-related internal representations within LLMs. By comparing the internal representations before and after fine-tuning, we figured out that using more length specific prompts, particularly  the ``Priming’’ prompt, causes the top-k hidden units related to length control to be more highly activated than those in the No-constraint and Length prompt settings.
>
> We also aim to investigate how LLMs process and learn length constraints between explicit fine-tuning and zero-shot in-context learning. Our experimental results in Table 4 show that the same top-k activated length-related hidden units are involved in both the zero-shot and fine-tuning settings when using the more length-specific ``Priming’’ prompt. The consistency in the highly activated units in both settings implies that LLMs interpret in-context learning as a form of implicit fine-tuning.
>
>
>
> Evaluation methodology unclear
> > How were models sampled? E.g. greedy decoding, top-k?
>
> We used greedy decoding for all settings in our experiments to avoid any randomness in the generation process.
>
> > As LMs have variable-length inputs, which tokens' representations are being used for the linear regressions?
>
> We apologize for any confusion. In our experiments, we used the representations of the generated output tokens, excluding the input tokens from the prompts. During the summary generation process, the model produces one token at each time step t. For each generated token, we extracted hidden states from the layers of LLM and assigned corresponding time steps.
>
> > l402: Please include a summary of how delta CR should be interpreted; please define compression ratio in the text.
>
> The compression ratio is calculated as the ratio of the length of the generated summary to the length of the source text. The length is counted based on the number of words.
>
> Delta CR is an arithmetical difference between ratios. Delta CR value close to zero indicates that the model-generated summary has a compression ratio similar to the gold summary. The more positive delta CR means the generated summary is longer than the gold summary, while a negative delta CR means it is shorter than the gold summary.
>
>
> > Table 4: "The numbers in parentheses indicate an index of hidden units from the second layer of the attention mechanisms." I did not understand this; how many hidden units are there total? Are the reported values for a single hidden unit, or averaged over several individual regressions?
>
> In Llama-2-13b-chat, the total hidden units from the layer is 5,120. Thus, we additionally trained classifiers for single hidden units. The reported values are for a single hidden unit.
>
> >Partial results reported in manuscript While Section 3 (Methods) states that experiments are on Llama 2, Llama 3, and Phi, only Section 4 reports results for all models. The rest of the paper only reports results on Llama 2. Please include the full set of results for Section 5.
>
> We appreciate your careful reading and valuable feedback, which have helped improve our draft. We investigated other models and observed that our method for identifying length-related units effectively controls the output sequence length. We will submit the revised draft, incorporating your feedback. Thank you very much.

---

> > ### Author Response · Authors · 2024-11-16
> > **Response to questions and weakness.**
> >
> > >No statistical significance / random seeds reported The lack of significance reporting makes it very difficult to quantitatively evaluate the results. In particular:
> > Table 6 (human evaluations). It is impossible to tell whether the differences between -10 and 10 are statistically significant without errors. Please include errors and significance values.
> >
> > In table 6, the difference between -10 and 10 are statistically significant based on paired-bootstrap-resampling with 10,000 random samples [2]. We will add daggers and standard deviations in the draft.
> >
> > Table 6
> > | **Scale** |      |  **Zero-shot**       |       |    |   **Fine-tuning**     |       |
> > |-----------|----------------------|-------|-------|--------------------|-------|-------|
> > |           | **Coh.**            | **Conc.** | **Infor.** | **Coh.**         | **Conc.** | **Infor.** |
> > | -10       | 3.73 (0.25)         | 3.56 (0.25) | **3.71**$^\dagger$ (0.27) | 3.43 (0.25) | _3.33_ (0.27) | **3.34**$^\dagger$ (0.21) |
> > | 1         | 3.72 (0.29)         | 3.59 (0.23) | 3.70 (0.29) | 3.48 (0.23) | 3.46 (0.20) | 3.31 (0.27) |
> > | Gold      | 3.67 (0.30)         | 3.58 (0.23) | 3.68 (0.27) | 3.42 (0.26) | 3.45 (0.22) | 3.28 (0.22) |
> > | 10        | 3.69 (0.24)         | 3.59 (0.29) | _3.63_ (0.29) | 3.41 (0.22) | **3.47**$^\dagger$ (0.22) | _3.19_ (0.23) |
> >
> >
> > [2] Philipp Koehn. 2004. Statistical significance tests for machine translation evaluation. In Proceedings of the 2004 Conference on Empirical Methods in Natural Language Processing. Association for Computational Linguistics.

---

> > > ### Author Response · Authors · 2024-11-16
> > > **Response to questions and weakness.**
> > >
> > > >Table 5: N is reported in the # data row, but are the values an average over N? If so, can you please report the standard deviation?
> > >
> > > Your understanding is correct. The values are average. We have added the standard deviation. The following is Table 5 with the standard deviation. Compared to Base which does not multiply factors, scaling +10 or -10 successfully generates shorter or longer summaries. Dagger indicates the statistical significance (p<0.05) using paired boot-strapping methods with 10,000 samples.
> > >
> > > | **Setting**    | **Prompting**   | **Scale** | **1-5**      | **6-10**     | **11-15**     | **16-20**     | **21-25**     | **25-30**     | **30-**      |
> > > |----------------|-----------------|-----------|--------------|--------------|---------------|---------------|---------------|---------------|--------------|
> > > | **Zero-shot**  | **Length**      | **#Data** | 2            | 91           | 275           | 282           | 177           | 102           | 71           |
> > > |                |            | **Base**  | -15.31 (15.3) / 50.00 (50.0) | 4.65 (15.8) / 78.30 (22.1) | 19.28 (17.06) / 73.64 (17.1) | 33.53 (15.7) / 65.53 (14.8) | 45.37 (15.1) / 56.29 (13.8) | 53.31 (12.2) / 49.39 (13.2) | 59.19 (11.5) / 43.93 (13.4) |
> > > |                |                 | **10**    | -15.31 (15.3) / 50.00 (50.0) | 10.38 (20.1) / 76.54 (21.7) | 22.44 (18.0) / 73.25 (16.7) | 34.53 (18.9) / 65.01 (16.0) | 46.81 (16.9) / 55.70 (14.4) | 52.71 (15.1) / 50.21 (14.6) | 60.26 (11.0) / 44.04 (13.5) |
> > > |                |                 | **-10**   | -15.31 (15.3) / 50.00 (50.0) | 20.62 (23.7)$^\dagger$ / 73.45 (20.7) | 33.58 (19.9)$^\dagger$ / 67.68 (16.5) | 43.09 (20.6)$^\dagger$ / 62.33 (16.6) | 55.65 (15.4)$^\dagger$ / 53.23 (14.1) | 55.93 (17.2)$^\dagger$ / 48.78 (14.5) | 63.98 (13.9)$^\dagger$ / 43.30 (13.3) |
> > > |                | **Priming**     | **#Data** | 33           | 324          | 382           | 176           | 57            | 24            | 4            |
> > > |                |                 | **Base**  | 0.71 (5.6) / 79.35 (25.3) | 3.22 (10.1) / 74.33 (24.3) | 11.82 (11.9) / 74.02 (18.6) | 22.03 (13.5) / 69.38 (17.4) | 35.66 (14.0) / 61.83 (15.0) | 39.35 (14.5) / 57.24 (13.1) | 44.90 (4.7) / 53.52 (17.8) |
> > > |                |                 | **10**    | 0.81 (5.4) / 80.65 (24.6) | 1.58 (10.6)$^\dagger$ / 74.79 (23.7) | 9.89 (15.1)$^\dagger$ / 73.92 (19.6) | 17.30 (18.9)$^\dagger$ / 71.49 (18.6)$^\dagger$ | 24.60 (21.1)$^\dagger$ / 64.60 (21.0) | 36.13 (19.0) / 60.77 (14.8) | 36.69 (20.4) / 63.91 (23.5) |
> > > |                |                 | **-10**   | 1.36 (13.0) / 61.89 (37.1) | 13.88 (22.7)$^\dagger$ / 66.25 (26.8) | 24.06 (24.4)$^\dagger$ / 64.38 (23.0) | 30.78 (23.2)$^\dagger$ / 63.02 (20.9) | 42.78 (20.5)$^\dagger$ / 58.57 (16.6) | 39.48 (22.1) / 57.42 (17.5) | 50.29 (8.3) / 56.60 (9.5) |
> > > | **Fine-tuning**| **Length**      | **#Data** | 28           | 464          | 417           | 82            | 7             | 2             | 0            |
> > > |                |                 | **Base**  | -2.12 (4.9) / 79.17 (30.1) | 1.34 (9.5) / 85.07 (18.1) | 6.97 (10.9) / 82.99 (17.1) | 13.01 (13.3) / 83.63 (12.5) | 19.85 (8.2) / 84.08 (7.7) | 38.65 (10.4) / 57.05 (26.3) | --           |
> > > |                |                 | **10**    | -1.68 (4.9) / 80.83 (29.3) | 1.26 (9.3) / 85.10 (18.1) | 4.41 (10.6)$^\dagger$ / 83.58 (17.9) | 5.42 (14.0)$^\dagger$ / 82.01 (16.9) | 10.37 (7.3) / 82.47 (20.6) | 26.83 (1.4) / 61.67 (21.7) | --           |
> > > |                |                 | **-10**   | 0.08 (8.5)$^\dagger$ / 73.22 (30.2) | 3.69 (11.8)$^\dagger$ / 83.26 (17.9) | 9.06 (12.4)$^\dagger$ / 82.45 (16.8) | 11.51 (14.7) / 81.52 (15.0) | 18.59 (9.5) / 85.07 (8.5) | 40.47 (12.3) / 58.74 (24.6) | --           |
> > > |                | **Priming**     | **#Data** | 64           | 565          | 321           | 49            | 1             | 0             | 0            |
> > > |                |                 | **Base**  | 0.07 (2.7) / 87.98 (25.0) | -0.41 (4.2) / 86.36 (20.0) | 0.06 (5.6) / 83.86 (18.6) | 1.00 (6.0) / 83.59 (15.3) | 0.00 (0.0) / 83.72 (0.0) | --           | --           |
> > > |                |                 | **10**    | -0.65 (3.5)$^\dagger$ / 87.47 (25.8) | -3.18 (5.1)$^\dagger$ / 84.86 (19.8) | -5.29 (6.7)$^\dagger$ / 81.38 (17.9) | -7.71 (6.7)$^\dagger$ / 80.34 (15.8) | -8.70 (0.0)$^\dagger$ / 78.05 (0.0) | --           | --           |
> > > |                |                 | **-10**   | 3.21 (5.2)$^\dagger$ / 81.36 (24.7) | 9.78 (8.8)$^\dagger$ / 78.59 (18.3) | 17.29 (9.7)$^\dagger$ / 77.47 (13.4) | 16.92 (10.2)$^\dagger$ / 79.22 (12.3) | 15.22 (0.0)$^\dagger$ / 84.00 (0.0) | --           | --           |

---

> > > > ### Author Response · Authors · 2024-11-16
> > > > **Response to questions and weakness.**
> > > >
> > > > >Figures: please report error bars with standard deviations over random seeds.
> > > > For all Tables and Figures, we will add error bars with standard deviations in the draft. The followings are Table 2 and Table 6.
> > > >
> > > >
> > > > The following is Table 2 including standard deviation. Please note that standard deviations for the numbers in Table 3 are also nearly zero.
> > > >
> > > >  **Model**                  | **Constraint**   | **Attn Out (F)** | **Attn Out (S)** | **Attn Out (L)** | **Attn Residual (F)** | **Attn Residual (S)** | **Attn Residual (L)** | **MLP Out (F)** | **MLP Out (S)** | **MLP Out (L)** | **MLP Residual (F)** | **MLP Residual (S)** | **MLP Residual (L)** |
> > > > |----------------------------|------------------|------------------|------------------|------------------|------------------------|-----------------------|-----------------------|-----------------|-----------------|-----------------|-----------------------|-----------------------|-----------------------|
> > > > | **Llama-2-7B**            | **No-constraint** | 0.94 (0.00)     | **0.95 (0.00)**  | 0.88 (0.00)      | 0.00 (0.00)           | 0.90 (0.00)          | 0.89 (0.00)          | 0.84 (0.00)    | 0.70 (0.00)    | 0.67 (0.01)    | 0.90 (0.00)           | 0.94 (0.00)           | 0.85 (0.00)          |
> > > > |                            | **Length**       | 0.98 (0.00)     | **0.99 (0.00)**  | 0.93 (0.00)      | 0.11 (0.00)           | 0.94 (0.00)          | 0.93 (0.00)          | 0.89 (0.00)    | 0.77 (0.00)    | 0.70 (0.01)    | 0.95 (0.00)           | 0.97 (0.00)           | 0.89 (0.00)          |
> > > > |                            | **Priming**      | 0.98 (0.00)     | **0.99 (0.00)**  | 0.95 (0.00)      | 0.11 (0.00)           | 0.94 (0.00)          | 0.94 (0.00)          | 0.89 (0.00)    | 0.77 (0.00)    | 0.78 (0.00)    | 0.95 (0.00)           | 0.98 (0.00)           | 0.92 (0.00)          |
> > > > | **Llama-2-13B**           | **No-constraint** | 0.95 (0.00)     | **0.96 (0.00)**  | 0.93 (0.00)      | 0.08 (0.01)           | 0.93 (0.00)          | 0.92 (0.00)          | 0.90 (0.00)    | 0.83 (0.00)    | 0.74 (0.01)    | 0.93 (0.00)           | 0.95 (0.00)           | 0.89 (0.00)          |
> > > > |                            | **Length**       | **0.94 (0.00)** | **0.94 (0.00)**  | 0.92 (0.00)      | 0.10 (0.00)           | 0.92 (0.00)          | 0.92 (0.00)          | 0.89 (0.00)    | 0.81 (0.00)    | 0.75 (0.00)    | 0.91 (0.00)           | **0.94 (0.00)**       | 0.91 (0.00)          |
> > > > |                            | **Priming**      | **0.99 (0.00)** | **0.99 (0.00)**  | 0.91 (0.00)      | 0.17 (0.00)           | 0.96 (0.00)          | 0.92 (0.00)          | 0.92 (0.00)    | 0.81 (0.00)    | 0.72 (0.00)    | 0.97 (0.00)           | 0.98 (0.00)           | 0.89 (0.00)          |
> > > > | **Llama-2-70B**           | **No-constraint** | 0.97 (0.00)     | **0.99 (0.00)**  | 0.95 (0.00)      | 0.16 (0.00)           | 0.93 (0.00)          | 0.92 (0.00)          | 0.83 (0.01)    | 0.81 (0.01)    | 0.82 (0.00)    | 0.95 (0.00)           | 0.98 (0.00)           | 0.92 (0.00)          |
> > > > |                            | **Length**       | 0.97 (0.00)     | **0.99 (0.00)**  | 0.94 (0.00)      | 0.17 (0.00)           | 0.92 (0.00)          | 0.93 (0.00)          | 0.87 (0.00)    | 0.84 (0.00)    | 0.80 (0.01)    | 0.95 (0.00)           | 0.98 (0.00)           | 0.92 (0.00)          |
> > > > |                            | **Priming**      | **0.98 (0.00)** | 0.97 (0.00)      | 0.91 (0.00)      | 0.18 (0.00)           | 0.91 (0.00)          | 0.89 (0.00)          | 0.82 (0.00)    | 0.76 (0.01)    | 0.78 (0.01)    | 0.94 (0.00)           | 0.95 (0.00)           | 0.88 (0.00)          |
> > > > | **Llama-3-8B**            | **No-constraint** | 0.96 (0.00)     | **0.98 (0.00)**  | 0.91 (0.00)      | 0.20 (0.00)           | 0.86 (0.00)          | 0.91 (0.00)          | 0.70 (0.00)    | 0.74 (0.00)    | 0.78 (0.01)    | 0.88 (0.00)           | 0.95 (0.00)           | 0.88 (0.00)          |
> > > > |                            | **Length**       | 0.96 (0.00)     | **0.97 (0.00)**  | 0.93 (0.00)      | 0.16 (0.00)           | 0.88 (0.00)          | 0.93 (0.00)          | 0.72 (0.00)    | 0.75 (0.00)    | 0.79 (0.01)    | 0.90 (0.00)           | 0.96 (0.00)           | 0.89 (0.00)          |
> > > > |                            | **Priming**      | 0.97 (0.00)     | **0.98 (0.00)**  | 0.94 (0.00)      | 0.24 (0.00)           | 0.87 (0.00)          | 0.94 (0.00)          | 0.73 (0.00)    | 0.76 (0.00)    | 0.87 (0.00)    | 0.89 (0.00)           | 0.95 (0.00)           | 0.92 (0.00)          |

---

> > > > > ### Author Response · Authors · 2024-11-16
> > > > >
> > > > > **Model**                  | **Constraint**   | **Attn Out (F)** | **Attn Out (S)** | **Attn Out (L)** | **Attn Residual (F)** | **Attn Residual (S)** | **Attn Residual (L)** | **MLP Out (F)** | **MLP Out (S)** | **MLP Out (L)** | **MLP Residual (F)** | **MLP Residual (S)** | **MLP Residual (L)** |
> > > > > |----------------------------|------------------|------------------|------------------|------------------|------------------------|-----------------------|-----------------------|-----------------|-----------------|-----------------|-----------------------|-----------------------|-----------------------|
> > > > > | **Phi3-mini-4k**          | **No-constraint** | 0.93 (0.00)     | **0.97 (0.00)**  | 0.91 (0.00)      | 0.07 (0.01)           | 0.80 (0.01)          | 0.91 (0.00)          | 0.61 (0.01)    | 0.66 (0.01)    | 0.55 (0.01)    | 0.84 (0.00)           | 0.95 (0.00)           | 0.86 (0.00)          |
> > > > > |                            | **Length**       | 0.94 (0.00)     | **0.97 (0.00)**  | 0.92 (0.00)      | 0.04 (0.00)           | 0.80 (0.00)          | 0.92 (0.00)          | 0.65 (0.01)    | 0.67 (0.00)    | 0.56 (0.01)    | 0.82 (0.00)           | 0.94 (0.00)           | 0.86 (0.00)          |
> > > > > |                            | **Priming**      | 0.93 (0.00)     | **0.97 (0.00)**  | 0.89 (0.00)      | 0.07 (0.01)           | 0.77 (0.00)          | 0.90 (0.00)          | 0.48 (0.01)    | 0.63 (0.01)    | 0.58 (0.01)    | 0.80 (0.00)           | 0.95 (0.00)           | 0.84 (0.00)          |
> > > > > | **Phi3-small-8k**         | **No-constraint** | 0.92 (0.01)     | **0.95 (0.00)**  | 0.71 (0.01)      | 0.01 (0.01)           | 0.83 (0.00)          | 0.83 (0.00)          | 0.76 (0.01)    | 0.72 (0.01)    | 0.48 (0.00)    | 0.87 (0.00)           | 0.90 (0.00)           | 0.81 (0.01)          |
> > > > > |                            | **Length**       | 0.94 (0.00)     | **0.97 (0.00)**  | 0.85 (0.00)      | 0.11 (0.01)           | 0.86 (0.00)          | 0.87 (0.00)          | 0.80 (0.00)    | 0.79 (0.01)    | 0.54 (0.01)    | 0.89 (0.00)           | 0.94 (0.00)           | 0.87 (0.00)          |
> > > > > |                            | **Priming**      | 0.97 (0.00)     | **0.98 (0.00)**  | 0.81 (0.01)      | 0.27 (0.01)           | 0.92 (0.00)          | 0.87 (0.01)          | 0.86 (0.01)    | 0.83 (0.00)    | 0.58 (0.00)    | 0.93 (0.00)           | 0.97 (0.00)           | 0.86 (0.00)          |

---

> > > > > > ### Author Response · Authors · 2024-11-16
> > > > > > **Response to questions and weakness.**
> > > > > >
> > > > > > We have conducted significant tests to determine whether the differences in conciseness and informativeness scores are statistically significant. We used paired-boot strap resampling with 10,000 samples [1]. We obtained the differences in conciseness and informativeness scores between scaling factors of -10 and +10 are statistically significant (p <0.05). We think this demonstrates that length-related units we found can directly control output sequence length to produce longer summaries with negative scalings or shorter summaries with positive scalings without external prompt changes.
> > > > > >
> > > > > > [1] Philipp Koehn. 2004. Statistical significance tests for machine translation evaluation. In Proceedings of the 2004 Conference on Empirical Methods in Natural Language Processing. Association for Computational Linguistics.
> > > > > >
> > > > > > Table 6
> > > > > > | **Scale** |      |  **Zero-shot**       |       |    |   **Fine-tuning**     |       |
> > > > > > |-----------|----------------------|-------|-------|--------------------|-------|-------|
> > > > > > |           | **Coh.**            | **Conc.** | **Infor.** | **Coh.**         | **Conc.** | **Infor.** |
> > > > > > | -10       | 3.73 (0.25)         | 3.56 (0.25) | **3.71**$^\dagger$ (0.27) | 3.43 (0.25) | _3.33_ (0.27) | **3.34**$^\dagger$ (0.21) |
> > > > > > | 1         | 3.72 (0.29)         | 3.59 (0.23) | 3.70 (0.29) | 3.48 (0.23) | 3.46 (0.20) | 3.31 (0.27) |
> > > > > > | Gold      | 3.67 (0.30)         | 3.58 (0.23) | 3.68 (0.27) | 3.42 (0.26) | 3.45 (0.22) | 3.28 (0.22) |
> > > > > > | 10        | 3.69 (0.24)         | 3.59 (0.29) | _3.63_ (0.29) | 3.41 (0.22) | **3.47**$^\dagger$ (0.22) | _3.19_ (0.23) |

---

> > > > > > > ### Comment · Reviewer_Khqj · 2024-11-30
> > > > > > >
> > > > > > > Thanks for your response and sorry for my late reply. The overwhelming majority of my clarification questions have been addressed, but some still remain.
> > > > > > >
> > > > > > > > As LMs have variable-length inputs, which tokens' representations are being used for the linear regressions?
> > > > > > > > In our experiments, we used the representations of the generated output tokens, excluding the input tokens from the prompts. During the summary generation process, the model produces one token at each time step t. For each generated token, we extracted hidden states from the layers of LLM and assigned corresponding time steps.
> > > > > > >
> > > > > > > I still don't understand. At each layer, you have $L$ outputs $X_l^{n \times d_{\text{model}}}$, $l=1\cdots L$, where $L$ is the number of tokens in the input to the Transformer. For the linear regression in line 190, do you take $\mathbf X_{l=L}$, which is the last-token representation?
> > > > > > >
> > > > > > > > Partial results reported in manuscript + No statistical significance / random seeds reported
> > > > > > >
> > > > > > > Thanks for adding the new results; I checked the new Figure 3 which includes all the new models. However, I'm not sure how to interpret them with the large error bars-- the figure is quite hard to read, and it seems like none of the trends are clear, as the other reviewers have suggested.
> > > > > > >
> > > > > > > In general, the large errors issue is present throughout the results, which I'm afraid leaves me still unconvinced... I'll maintain my recommendation to reject the paper. Thanks so much for the discussion, and I believe the new results are a step in the right direction.

---

### Official Review · Reviewer_Xgdv · 2024-11-07

**Soundness:** 2
**Presentation:** 2
**Contribution:** 2
**Rating:** 5
**Confidence:** 2

**Summary:**

The paper describes experiments into controlling LLM output sequence length in a sentence summarization task; it finds that particular units correlate with output length, and that adjusting their weights gives a way of controlling output length. This is evaluated in terms of the resulting length compression ratios and content recall (measured via ROUGE), and also with some human evaluations.

**Strengths:**

The focus on internal mechanisms and the effect of adjusting them seems a nice contribution in terms of understanding how LLMs can be effectively controlled.

The results suggest that this method may allow length to be controlled more directly than with other methods, and may be able to do so without too much effect on output quality.

**Weaknesses:**

The evaluation and comparison are hard to follow (for me at least), and it is not easy to see how the results of this method compare to the results that would be achieved using other methods (some of which are mentioned in Section 2). It is hard in places to understand how the results displayed in the tables and figures correspond to the way they are summarized in the text.

I found some key details hard to understand; for example, the use of fine-tuning is important here, with experiments comparing results with & without fine-tuning, but I couldn't find any explanation of what the fine-tuning objective was; one of the key metrics (delta-CR) is explained at a high level but not in detail, so it is hard to know what the numbers really mean and how to compare them.

**Questions:**

What fine-tuning objective is actually being used here - is the fine-tuning supervised by output sequence length? (Apologies if these details are given - I did try to find them but may have missed something).

How is delta-CR calculated - is it an arithmetical difference between ratios, or e.g. a log ratio of ratios? Are numbers closer to zero better in principle? Is the same gold-standard length used as the norm even when the multiplication factor varies - and would that mean that numbers closer to zero aren't actually "better" if the multiplication factor is being set e.g. to shorten or lengthen outputs?

How should we compare the results in terms of delta-CR and ROUGE compared to alternative approaches e.g. the length priming approach? The "base" / "Priming" entries in tables are presumably representative of that approach, but I wasn't clear whether the prompt is then varied to produce different length outputs, in the same way that the multiplication factor is being changed, so that it's a fair comparison.

On page 8 the discussion says that "R-L scores slightly decrease when the No- constraint and Length prompts were used. In comparison, for Priming, which is more length-specific prompts, continues to improve performance even when we applied a large scaling factor of 10". I guess this means "decrease relative to Base", but this doesn't quite seem to fit with the graphs (Figure 2a), where the No-constraint and Length prompts show some increases in R-L, and for Priming there are some large drops in R-L at the other end of the range. Am I interpreting the graphs correctly?

The caption of Table 5 talks about "Results based on word length" but I guess it means "output length (#words)" rather than actually word length (i.e. number of characters per word).

---

> ### Author Response · Authors · 2024-11-16
> **Response to questions and weakness.**
>
> > What fine-tuning objective is actually being used here - is the fine-tuning supervised by output sequence length?
>
> The models were fine-tuned on the Google sentence summarization dataset, which consists of instruction-based formats, No-constraint, Length, Priming in Table 1 [1]. The fine-tuning process is supervised by incorporating length-specific instructions to enhance the model’s ability to control the output sequence length.
>
> [1] Do, 2024. InstructCMP: Length Control in Sentence Compression through Instruction-based Large Language Models, Findings of the Association for Computational Linguistics: ACL 2024.
>
> > How is delta-CR calculated - is it an arithmetical difference between ratios, or e.g. a log ratio of ratios? Are numbers closer to zero better in principle?
>
> The compression ratio is calculated as the ratio of the length of the generated summary to the length of the source text. The length is counted based on the number of words. Delta CR is an arithmetical difference between ratios.
> Delta CR value close to zero indicates that the model-generated summary has a compression ratio similar to the gold summary. The more positive delta CR means the generated summary is longer than the gold summary, while a negative delta CR means it is shorter than the gold summary.
>
> > Is the same gold-standard length used as the norm even when the multiplication factor varies - and would that mean that numbers closer to zero aren't actually "better" if the multiplication factor is being set e.g. to shorten or lengthen outputs?
>
> Your understanding is correct. The same gold-standard length is used as the norm even when the multiplication factor varies in order to demonstrate that the length-related units are actually involved in length-control during generation process. If the goal is to generate summaries that meet the gold summary’s length, then delta CR closer to zero is better. However, when we multiply scaling factors, the gold summary length does not change. Thus, if we aim to generate longer summaries, an increased delta CR indicates success compared to not multiplying scaling factors. If we aim to generate shorter summaries, a decreased delta CR indicates success compared to not multiplying scaling factors.
>
> > How should we compare the results in terms of delta-CR and ROUGE compared to alternative approaches e.g. the length priming approach?
>
> Our goal is to demonstrate that we can control the output sequence length by identifying and manipulating length-related units within LLMs. By scaling those units, we aim to show that they indeed have a direct impact on output length during the generation process without changing the prompt. "Base" indicates we did not scale length-related units and solely generated summaries with the given prompts, such as "priming". Thus, we maintained consistent prompts between "priming" and scaling length-related units for fair comparisons. By comparing delta CR and ROUGE scores under consistent settings, we show that multiplication factors to the length-related top-1 unit even control output length without losing informativeness.
>
> The following is Table 5 with the standard deviation. Compared to Base which does not multiply factors, scaling +10 or -10 successfully generates shorter or longer summaries. Dagger indicates the statistical significance (p<0.05) using paired boot-strapping methods with 10,000 samples [1].
>
> [1] Philipp Koehn. 2004. Statistical significance tests for machine translation evaluation. In Proceedings of the 2004 Conference on Empirical Methods in Natural Language Processing. Association for Computational Linguistics.

---

> > ### Author Response · Authors · 2024-11-16
> > **Response to questions and weakness.**
> >
> > | **Setting**    | **Prompting**   | **Scale** | **1-5**      | **6-10**     | **11-15**     | **16-20**     | **21-25**     | **25-30**     | **30-**      |
> > |----------------|-----------------|-----------|--------------|--------------|---------------|---------------|---------------|---------------|--------------|
> > | **Zero-shot**  | **Length**      | **#Data** | 2            | 91           | 275           | 282           | 177           | 102           | 71           |
> > |                |            | **Base**  | -15.31 (15.3) / 50.00 (50.0) | 4.65 (15.8) / 78.30 (22.1) | 19.28 (17.06) / 73.64 (17.1) | 33.53 (15.7) / 65.53 (14.8) | 45.37 (15.1) / 56.29 (13.8) | 53.31 (12.2) / 49.39 (13.2) | 59.19 (11.5) / 43.93 (13.4) |
> > |                |                 | **10**    | -15.31 (15.3) / 50.00 (50.0) | 10.38 (20.1) / 76.54 (21.7) | 22.44 (18.0) / 73.25 (16.7) | 34.53 (18.9) / 65.01 (16.0) | 46.81 (16.9) / 55.70 (14.4) | 52.71 (15.1) / 50.21 (14.6) | 60.26 (11.0) / 44.04 (13.5) |
> > |                |                 | **-10**   | -15.31 (15.3) / 50.00 (50.0) | 20.62 (23.7)$^\dagger$ / 73.45 (20.7) | 33.58 (19.9)$^\dagger$ / 67.68 (16.5) | 43.09 (20.6)$^\dagger$ / 62.33 (16.6) | 55.65 (15.4)$^\dagger$ / 53.23 (14.1) | 55.93 (17.2)$^\dagger$ / 48.78 (14.5) | 63.98 (13.9)$^\dagger$ / 43.30 (13.3) |
> > |                | **Priming**     | **#Data** | 33           | 324          | 382           | 176           | 57            | 24            | 4            |
> > |                |                 | **Base**  | 0.71 (5.6) / 79.35 (25.3) | 3.22 (10.1) / 74.33 (24.3) | 11.82 (11.9) / 74.02 (18.6) | 22.03 (13.5) / 69.38 (17.4) | 35.66 (14.0) / 61.83 (15.0) | 39.35 (14.5) / 57.24 (13.1) | 44.90 (4.7) / 53.52 (17.8) |
> > |                |                 | **10**    | 0.81 (5.4) / 80.65 (24.6) | 1.58 (10.6)$^\dagger$ / 74.79 (23.7) | 9.89 (15.1)$^\dagger$ / 73.92 (19.6) | 17.30 (18.9)$^\dagger$ / 71.49 (18.6)$^\dagger$ | 24.60 (21.1)$^\dagger$ / 64.60 (21.0) | 36.13 (19.0) / 60.77 (14.8) | 36.69 (20.4) / 63.91 (23.5) |
> > |                |                 | **-10**   | 1.36 (13.0) / 61.89 (37.1) | 13.88 (22.7)$^\dagger$ / 66.25 (26.8) | 24.06 (24.4)$^\dagger$ / 64.38 (23.0) | 30.78 (23.2)$^\dagger$ / 63.02 (20.9) | 42.78 (20.5)$^\dagger$ / 58.57 (16.6) | 39.48 (22.1) / 57.42 (17.5) | 50.29 (8.3) / 56.60 (9.5) |
> > | **Fine-tuning**| **Length**      | **#Data** | 28           | 464          | 417           | 82            | 7             | 2             | 0            |
> > |                |                 | **Base**  | -2.12 (4.9) / 79.17 (30.1) | 1.34 (9.5) / 85.07 (18.1) | 6.97 (10.9) / 82.99 (17.1) | 13.01 (13.3) / 83.63 (12.5) | 19.85 (8.2) / 84.08 (7.7) | 38.65 (10.4) / 57.05 (26.3) | --           |
> > |                |                 | **10**    | -1.68 (4.9) / 80.83 (29.3) | 1.26 (9.3) / 85.10 (18.1) | 4.41 (10.6)$^\dagger$ / 83.58 (17.9) | 5.42 (14.0)$^\dagger$ / 82.01 (16.9) | 10.37 (7.3) / 82.47 (20.6) | 26.83 (1.4) / 61.67 (21.7) | --           |
> > |                |                 | **-10**   | 0.08 (8.5)$^\dagger$ / 73.22 (30.2) | 3.69 (11.8)$^\dagger$ / 83.26 (17.9) | 9.06 (12.4)$^\dagger$ / 82.45 (16.8) | 11.51 (14.7) / 81.52 (15.0) | 18.59 (9.5) / 85.07 (8.5) | 40.47 (12.3) / 58.74 (24.6) | --           |
> > |                | **Priming**     | **#Data** | 64           | 565          | 321           | 49            | 1             | 0             | 0            |
> > |                |                 | **Base**  | 0.07 (2.7) / 87.98 (25.0) | -0.41 (4.2) / 86.36 (20.0) | 0.06 (5.6) / 83.86 (18.6) | 1.00 (6.0) / 83.59 (15.3) | 0.00 (0.0) / 83.72 (0.0) | --           | --           |
> > |                |                 | **10**    | -0.65 (3.5)$^\dagger$ / 87.47 (25.8) | -3.18 (5.1)$^\dagger$ / 84.86 (19.8) | -5.29 (6.7)$^\dagger$ / 81.38 (17.9) | -7.71 (6.7)$^\dagger$ / 80.34 (15.8) | -8.70 (0.0)$^\dagger$ / 78.05 (0.0) | --           | --           |
> > |                |                 | **-10**   | 3.21 (5.2)$^\dagger$ / 81.36 (24.7) | 9.78 (8.8)$^\dagger$ / 78.59 (18.3) | 17.29 (9.7)$^\dagger$ / 77.47 (13.4) | 16.92 (10.2)$^\dagger$ / 79.22 (12.3) | 15.22 (0.0)$^\dagger$ / 84.00 (0.0) | --           | --           |

---

> > > ### Author Response · Authors · 2024-11-16
> > > **Response to questions and weakness.**
> > >
> > > > On page 8 the discussion says that "R-L scores slightly decrease when the No- constraint and Length prompts were used. In comparison, for Priming, which is more length-specific prompts, continues to improve performance even when we applied a large scaling factor of 10". I guess this means "decrease relative to Base", but this doesn't quite seem to fit with the graphs (Figure 2a), where the No-constraint and Length prompts show some increases in R-L, and for Priming there are some large drops in R-L at the other end of the range. Am I interpreting the graphs correctly?
> > >
> > > You are correct. It means "compared to Base". When we use the "Priming" prompt in the fine-tuned setting, we observe some drops in R-L when applying positive scaling factors such as +5 and +10 to the top-k highly activated length-related units. This is because the fine-tuned "Priming" prompt already precisely controls the output sequence length while generating informative summaries reported in Figure 2 (b) and the bin-wise analysis in Table 5. Thus, multiplying large positive scaling factors, which generate even shorter summaries, resulted in dropping R-L due to the potential gaps between gold and generated summaries. Even when applying large negative or positive scaling factors to the identified top-1 length-related units that generate shorter or longer summaries, the Rouge-L scores remain higher compared to those generated using the "No-constraint" and "Length" prompts.
> > >
> > > In contrast, for the "Length" prompt, large positive scaling factors affect the generated summaries closer to the gold summary length, thus, it improves performances.
> > >
> > >
> > > > The caption of Table 5 talks about "Results based on word length" but I guess it means "output length (#words)" rather than actually word length (i.e. number of characters per word).
> > >
> > > Your understanding is correct.

---

### Author Response · Authors · 2024-11-16
**We greatly appreciate your valuable reviews' comments.**

We greatly appreciate your valuable feedback to improve our draft.

We will revise and upload the improved version by incorporating all feedback as soon as possible. Please feel free to contact us if you have any further questions.

---

### Note · Authors · 2024-12-03

**Comment:**

We would like to appreciate all reviewers for their thorough reviews and comments. We hope to address these concerns in the next version.

**Withdrawal Confirmation:**

I have read and agree with the venue's withdrawal policy on behalf of myself and my co-authors.